# Intestinal microbiota-derived short-chain fatty acids regulation of immune cell IL-22 production and gut immunity

Wenjing Yang[1], Tianming Yu[1,2], Xiangsheng Huang[1], Anthony J. Bilotta[1], Leiqi Xu[1], Yao Lu[1], Jiaren Sun[1], Fan Pan[3], Jia Zhou [4], Wenbo Zhang[5], Suxia Yao[1], Craig L. Maynard [6], Nagendra Singh [7], Sara M. Dann [8], Zhanju Liu [2] & Yingzi Cong [1,9 ✉]

Innate lymphoid cells (ILCs) and CD4[+] T cells produce IL-22, which is critical for intestinal immunity. The microbiota is central to IL-22 production in the intestines; however, the factors that regulate IL-22 production by CD4[+] T cells and ILCs are not clear. Here, we show that microbiota-derived short-chain fatty acids (SCFAs) promote IL-22 production by CD4[+] T cells and ILCs through G-protein receptor 41 (GPR41) and inhibiting histone deacetylase (HDAC). SCFAs upregulate IL-22 production by promoting aryl hydrocarbon receptor (AhR) and hypoxia-inducible factor 1α (HIF1α) expression, which are differentially regulated by mTOR and Stat3. HIF1α binds directly to the *Il22* promoter, and SCFAs increase HIF1α binding to the *Il22* promoter through histone modification. SCFA supplementation enhances IL-22 production, which protects intestines from inflammation. SCFAs promote human CD4[+] T cell IL-22 production. These findings establish the roles of SCFAs in inducing IL-22 production in CD4[+] T cells and ILCs to maintain intestinal homeostasis.

[1] Department of Microbiology and Immunology, The University of Texas Medical Branch, Galveston, TX 77555, USA. [2] Department of Gastroenterology, The Shanghai Tenth People's Hospital, 200072 Shanghai, China. [3] Immunology and Hematopoiesis Division, Department of Oncology, Sidney Kimmel Comprehensive Cancer Center, Johns Hopkins University School of Medicine, Baltimore, MD 21287, USA. [4] Chemical Biology Program, Department of Pharmacology and Toxicology, The University of Texas Medical Branch, Galveston, TX 77555, USA. [5] Department of Ophthalmology and Visual Sciences, The University of Texas Medical Branch, Galveston, TX 77555, USA. [6] Department of Pathology, University of Alabama at Birmingham, Birmingham, AL 35294, USA. [7] Department of Biochemistry and Molecular Biology, Georgia Cancer Center, Augusta University, Augusta, GA 30912, USA. [8] Department of Internal Medicine, The University of Texas Medical Branch, Galveston, TX 77555, USA. [9] Department of Pathology, The University of Texas Medical Branch, Galveston, TX 77555, USA. ✉email: yicong@utmb.edu

nterleukin 22 (IL-22), a member of the IL-10 family, was initially characterized as a Th1 cytokine[1], and later as a Th17 as well as Th22 cytokine[2,3], which is central to host protection against inflammatory insult in the intestine by inducing antimicrobial peptides and promoting epithelial barrier function[4,5]. Both innate lymphoid cells (ILCs) and CD4[+] T cells produce IL-22. Although IL-22-producing ILCs express the transcription factors aryl hydrocarbon receptor (AhR) and the retinoid-related orphan receptor gamma t (RORγt), and require IL-23 stimulation to produce IL-22, a recent study demonstrated that CD4[+] T cells IL-22 production depends on AhR and T-bet, but not primarily IL-23, for IL-22 production[6]. IL-22 receptor complex, a heterodimer of IL-10R2 and IL-22R1, is confined to non-hematopoietic cells, and thus the IL-22-IL-22R axis provides a critical link in the integration of immune responses with barrier function at the mucosal surface[5,7]. Accumulating evidence indicates an important role of IL-22 in inflammatory bowel disease (IBD), in that the majority of IL-22-associated molecules are encoded by IBD susceptibility genes[8]. IL-22 produced by both innate and adaptive lymphocytes is indispensable for maintaining intestine homeostasis[9]. Although ILCs provide a rapid source of IL-22 that is essential for early protection of epithelial barrier function upon inflammatory insult, CD4[+] T cells become the dominant source of IL-22 in the intestinal lamina propria (LP) during chronic intestinal inflammation. However, the factors that regulate IL-22 production in CD4[+] T cells and ILCs in the intestines and the mechanisms involved are still not completely understood.

Gut microbiota is crucial in IL-22 production in the intestines as evidenced by the fact that germ-free mice show impaired IL-22 production[10–12]. However, how microbiota regulates IL-22 production is still unclear. AhR ligands produced by specific gut bacteria species are able to stimulate ILC3 and CD4[+] T cells to produce IL-22[13–16]. A recent study reported that *Clostridia* colonization of antibiotic-treated neonatal mice induces IL-22 production by ILCs and CD4[+] T cells[17]. Interestingly, *Clostridia* produce short-chain fatty acids (SCFAs)[18,19], the major metabolic products of gut microbiota from dietary fiber. SCFAs have been recognized as important mediators in maintaining intestinal homeostasis through regulating different cells[20–26]. Furthermore, G-protein-coupled receptor (GPR)43, one of the major receptors for SCFAs, has been recently reported to regulate ILC3 function[27]. However, whether and how SCFAs regulate IL-22 production in CD4[+] T cells and ILCs remains unknown.

In this report, we demonstrate that SCFAs promote IL-22 production in CD4[+] T cells and ILCs through histone deacetylase (HDAC) inhibition and GPR41, but not GPR43 and GPR109a. SCFAs upregulate IL-22 production through promoting AhR and hypoxia-inducible factor (HIF)1α expression, which are differentially regulated by mTOR and Stat3. HIF1α directly binds to the *Il22* promoter, and SCFAs increase the accessibility of HIF1α-binding sites in the *Il22* promoter through histone modification. Furthermore, SCFA supplementation in vivo protects mice from intestinal inflammation upon *Citrobacter rodentium* infection and inflammatory insult, which is mediated by enhanced IL-22 production.

## Result

**SCFAs promote IL-22 in CD4[+] T cells and ILCs in vitro.** SCFAs have been shown to promote regulatory T cell (Treg) development as well as CD4[+] T cell IL-10 production[20,26,28]. To explore how SCFAs regulate CD4[+] T cells more comprehensively, splenic CD4[+] T cells were isolated from wild-type (WT) C57BL/6J (B6) mice and activated with anti-CD3 mAb and anti-CD28 mAb in the presence or absence of butyrate, one of the major SCFAs in the intestines, for 2 days. The RNA transcriptome was analyzed by RNA sequencing (RNA-seq). Principal component analysis (PCA) and volcano plot analysis demonstrated butyrate treatment led to a different transcriptional profile (Supplementary Fig. 1a, b). Consistent with previous studies[20,28], butyrate promoted expression of *Il10* and *Ifng* by CD4[+] T cells (Fig. 1a). Interestingly, *Il22* was significantly increased in butyrate-treated CD4[+] T cells (Fig. 1a). To confirm SCFAs induction of IL-22 in gut microbiota antigen-specific T cells, we cultured splenic CD4[+] T cells of CBir1 TCR transgenic (CBir1 Tg) mice, which are specific for an immunodominant microbiota antigen CBir1 flagellin[29], with antigen-presenting cells (APCs) and CBir1 peptide in the presence or absence of acetate, propionate, or butyrate, the three major SCFAs, for 2 days. Acetate, propionate, and butyrate all increased IL-22 production at both mRNA and protein (Fig. 1b, c). We also confirmed that acetate, propionate, and butyrate increased *Il22* expression in WT B6 CD4[+] T cells activated with anti-CD3 mAb and anti-CD28 mAb (Supplementary Fig. 1c, d).

IL-22 was initially characterized as a Th1 cytokine, and later as a Th17 as well as Th22 cytokine[30]. To determine whether SCFAs promote IL-22 production in different T cell subtypes, CBir1 CD4[+] T cells were cultured under neutral, Th1, Th17, and Treg polarization conditions, in the presence or absence of butyrate, for 2 days. Butyrate induced *Il22* expression in CD4[+] T cells under all conditions (Fig. 1d), although Treg cells expressed the lowest *Il22* levels without butyrate stimulation among all CD4[+] T cell subtypes tested, which is consistent with the previous reports[31]. Interestingly, *Il22* expression in Th1 cells was higher even than those under Th17 conditions (Fig. 1d), which is likely due to TGFβ inhibition of IL-22 expression under Th17 conditions[32,33], and higher *Tbx21* expression in Th1 cells (Supplementary Fig. 1e), which is critical for inducing IL-22 in CD4[+] T cells[6,34]. We then performed RNA-seq to determine transcriptional profiles in CD4[+] T cells under Th1 conditions with or without butyrate treatment. Treatment with butyrate changed the transcription landscape of CD4[+] T cells under Th1 conditions (Supplementary Fig. 1f, g). Consistently, *Il22*, in addition to *Il10* and *Ifng*, was increased after the treatment of butyrate (Fig. 1e). We then measured *Il22* expression at different time points in CD4[+] T cell cultures with or without butyrate under Th1 conditions. Butyrate induced *Il22* expression as early as at 24 h and reached peak mRNA level at 60 h (Fig. 1f), indicating that butyrate promotes *Il22* expression in a time-dependent manner. Consistently, butyrate increased IL-22 protein in CD4[+] T cells under Th1 conditions (Fig. 1g, h). However, butyrate did not affect IFN-γ production in CD4[+] T cells (Fig. 1h and Supplementary Fig. 2). We obtained similar results in CD4[+] T cells cultured under Th17 conditions, in that butyrate promoted CD4[+] T cell production of IL-22 (Supplementary Fig. 1h, i). Butyrate also promoted Treg but inhibited Th17 cell differentiation (Supplementary Fig. 2). We then investigated whether SCFAs regulate IL-22 production in CD4[+] T cells from spleen and MLN differently. Similar to splenic CD4[+] T cells, butyrate treatment of MLN CD4[+] T cells enhanced IL-22 production (Supplementary Fig. 3a). Furthermore, splenic and MLN CD4[+] T cells proliferated (Supplementary Fig. 3b) and polarized at similar levels under Th1, Th17, and Treg conditions (Supplementary Fig. 3c).

Given that gut microbiota is required for IL-22 production in ILCs[12], we next investigated whether SCFAs regulate IL-22 in ILCs. CD4[+] T cell-depleted splenic cells were stimulated with IL-23 in the presence or absence of acetate, propionate, and butyrate overnight. Acetate, propionate, and butyrate upregulated IL-22 production, but not IL-17, in ILCs when cells were treated with IL-23 (Fig. 1i). However, SCFA treatment alone did not affect IL-22 production in ILCs without IL-23 stimulation (Supplementary

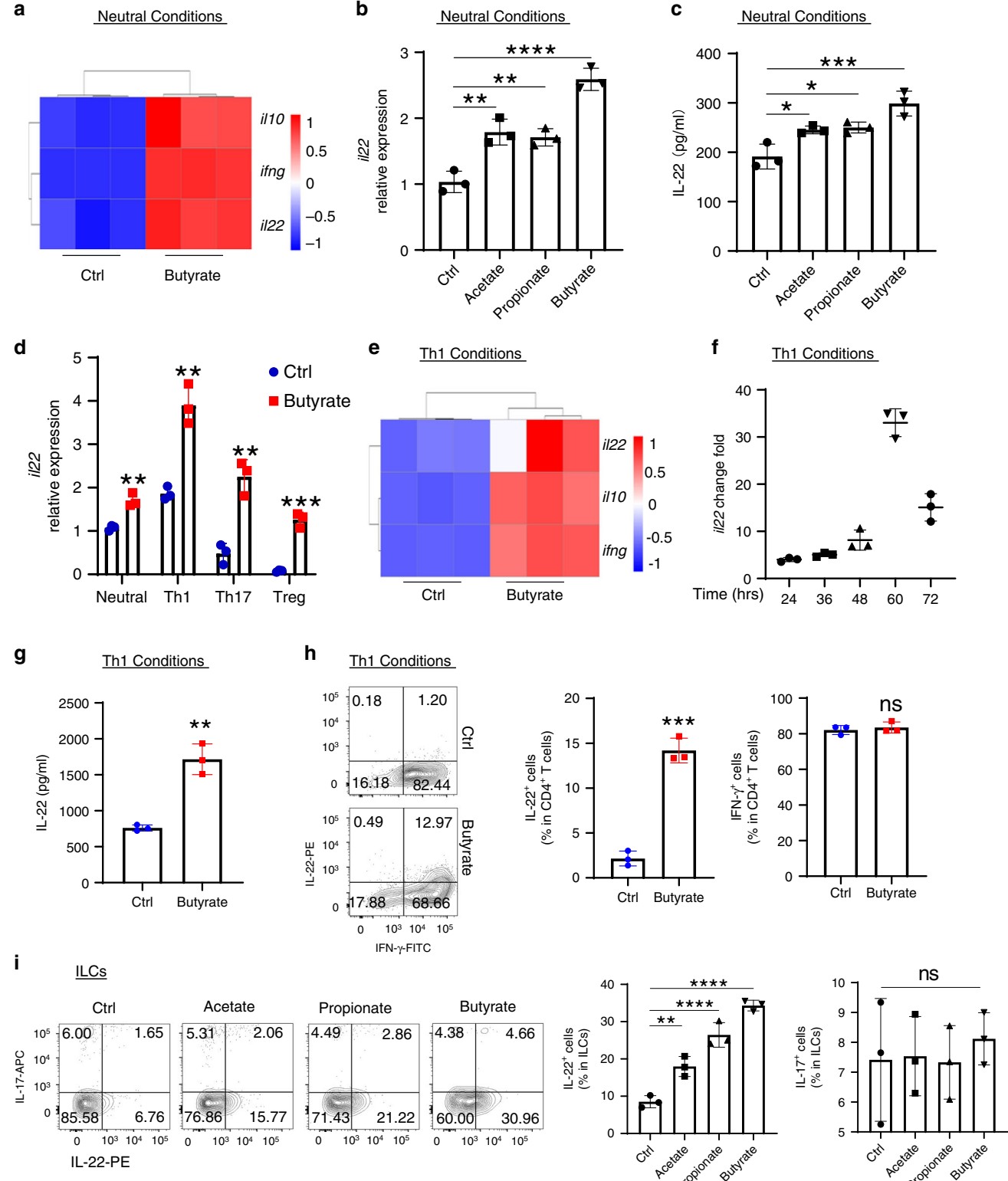

Fig. 4). Consistently, butyrate upregulated ILC production of IL-22 in LP (Supplementary Fig. 3d). Given IL-22 can be produced by NKT cells[35], we also determined whether butyrate affect NKT cell production of IL-22. However, butyrate did not affect IL-22 in NKT cells (Supplementary Fig. 5a).

**Butyrate promotes IL-22 production by CD4+ T cells and ILCs in vivo**. To investigate whether SCFAs upregulate IL-22 production

in vivo, WT B6 mice were administrated with or without 200 mM butyrate in drinking water for 3 weeks. Butyrate supplementation did not affect weight gain (Fig. 2a). Butyrate levels in the colon were increased after administration of butyrate (Fig. 2b). When mice were killed on day 21, butyrate upregulated IL-22 production in the serum and the colonic organ culture (Fig. 2c, d), and increased IL-22 production in CD4+ T cells in the spleen, MLN, and LP compared with control mice (Fig. 2e). Some CD4− cells

**Fig. 1 SCFAs promote IL-22 production in CD4$^+$ T cells and ILCs in vitro. a** WT splenic CD4$^+$ T cells were activated with anti-CD3/CD28 mAbs ± butyrate (0.5 mM) for 2 days ($n = 3$ biologically independent samples per group). RNA sequencing was performed. *Il10*, *Ifng*, and *Il22* expressions were shown in heatmap. **b, c** CBir1 Tg CD4$^+$ T cells were cultured with APCs and CBir1 peptide ± acetate (10 mM), propionate (0.5 mM), or butyrate (0.5 mM) for 2 days ($n = 3$/group). *Il22* expression was analyzed by qRT-PCR (**b**), and IL-22 in supernatants was assessed by ELISA (**c**). **d** CBir1 Tg CD4$^+$ T cells were cultured with APCs and CBir1 peptide ± butyrate (0.5 mM) for 2 days ($n = 3$/group) under neutral, Th1, Th17, or Treg conditions. *Il22* was analyzed by qRT-PCR. **e** CD4$^+$ T cells were activated with anti-CD3/CD28 mAbs ± butyrate (0.5 mM) for 2 days ($n = 3$ biologically independent samples per group) under Th1 conditions. RNA sequencing was performed. Expression of *Il10*, *Ifng*, and *Il22* was shown in heatmap. **f–h** CBir1 Tg CD4$^+$ T cells were activated with APCs and CBir1 peptide ± butyrate (0.5 mM) under Th1 conditions ($n = 3$/group). IL-22 was analyzed by qRT-PCR at different time point (**f**), and ELISA at 60 h (**g**), and IL-22 and IL-17 were measured flow cytometry on day 5 (**h**). **i** CD4$^+$ T cell-depleted splenic cells were treated with IL-23 (20 ng/ml) ± acetate (10 mM), propionate (0.5 mM), or butyrate (0.5 mM) for 16 h ($n = 3$/groups). IL-22 and IL-17 production in ILCs were analyzed by flow cytometry. One representative of three independent experiments was shown (**b–d**, **f–i**). Data were expressed as mean ± SD. Statistical significance was tested by two-tailed one-way ANOVA (**b, c, i**) or two-tailed unpaired Student *t*-test (**d, g, h**). **b** **p $= 0.0014$ (acetate vs control) and 0.0028 (propionate vs control), ****$p < 0.0001$; **c** *$p = 0.0212$ (acetate vs control) and 0.0139 (propionate vs control), ***$p < 0.0004$; **d** **$p = 0.0029$ (neutral), 0.0019 (Th1), and 0.0026 (Th17), ***$p = 0.0003$; **g** **$p = 0.0016$; **h** ***$p = 0.0002$, ns, no significance; **i** **$p = 0.0033$, ****$p < 0.0001$, ns no significance.

also expressed IL-22 (Supplementary Fig. 6). Butyrate also promoted ILC production of IL-22 in the spleen, MLN, and LP in vivo (Fig. 2g and Supplementary Fig. 7a, b). To investigate the expression of IL-22 in the different CD4$^+$ T cell subsets in vivo, IL-22 levels from LP IFN-γ$^+$ Th1, IL-17$^+$ Th17, Foxp3$^+$ Treg cells, and RORγt$^+$ ILC3 cells were analyzed. IL-22 was higher in Th1 and Th17 than Treg cells. Butyrate promoted IL-22 production in Th1, Th17, and RORγt$^+$ ILC3 cells, but not in Treg cells (Fig. 2f, h). IL-23 production in dendritic cells (DCs) is critical in regulating IL-22 production in ILCs[36]. However, butyrate did not affect IL-23p19 expression in LP DCs (Supplementary Fig. 5b). Taken all together, these data indicate that SCFAs promote IL-22 production in CD4$^+$ T cells and ILCs in vivo.

**Butyrate promotes IL-22 production through GPR41 and HDAC inhibition.** It has been shown that SCFAs function through binding their receptors, GPR43, GPR41, and GPR109a, and through inhibition of HDAC[37,38]. To determine whether GPR43, GPR41, and GPR109a mediate butyrate induction of IL-22 in CD4$^+$ T cells, we cultured CD4$^+$ T cells from WT, *Gpr43*$^{-/-}$ mice, and *Gpr109a*$^{-/-}$ mice, with or without butyrate under Th1 or Th17 conditions. IL-22 was induced at similar levels in WT and GPR43-deficient or GPR109a-deficient CD4$^+$ T cells under both Th1 and Th17 conditions (Supplementary Fig. 8a–h). However, treatment with GPR41-specific agonist, AR420626, promoted CD4$^+$ T cell IL-22 expression at both mRNA and protein levels under Th1 (Fig. 3a–c) and Th17 (Supplementary Fig. 8i–j) conditions, indicating that GPR41, but not GPR43 and GPR109a, mediates butyrate induction of IL-22 in CD4$^+$ T cells.

To investigate whether inhibition of HDAC also contributes to SCFA induction of CD4$^+$ T cell IL-22 production, we first determined whether butyrate at the dose of 0.5 mM could suppress HDAC activity. We treated CD4$^+$ T cells cultured under Th1 condition with or without butyrate for 24 h. Butyrate indeed inhibited HDAC activity in CD4$^+$ T cells (Fig. 3d). We then treated CD4$^+$ T cells cultured under Th1 or Th17 conditions with the HDAC inhibitor, Trichostatin A (TSA), to determine whether it mimics the effect of butyrate on the induction of IL-22. Treatment with TSA increased IL-22 production at both mRNA and protein levels in CD4$^+$ T cells under Th1 (Fig. 3a–c) and Th17 (Supplementary Fig. 8i–j) conditions. Furthermore, treatment with HDAC inhibitor and GPR41 agonist together further promoted IL-22 production in CD4$^+$ T cells to levels similar to that induced by butyrate (Fig. 3a–c). Butyrate induced similar levels of IL-22 in WT, *Gpr43*$^{-/-}$, and *Gpr109*$^{-/-}$ ILCs (Supplementary Fig. 9a, b). Similar to our findings in CD4$^+$ T cells, treatment with AR420626 or TSA promoted IL-22

production in ILCs (Supplementary Fig. 9c). Taken together, these data indicated that butyrate promotes IL-22 production through GPR41 and inhibiting HDAC in CD4$^+$ T cells and ILCs.

**HIF1α and AHR mediate IL-22 production induced by butyrate via GPR41.** We then investigated the mechanisms involved in butyrate induction of IL-22 production. RNA-seq data showed that *Ahr*, a transcription factor for IL-22 through directly binding to *Il22* promoter[14,39,40], and *Hif1a*, a transcription factor mediating hypoxia that has been recently shown to regulate IL-22 production in CD4$^+$ T cells[41], were upregulated in butyrate-treated CD4$^+$ T cells (Fig. 4a). Butyrate promotion of AhR and HIF1α expression in CD4$^+$ T cells under Th1 conditions was verified at both RNA and protein levels by qRT-PCR (Fig. 4b, c) and western blot (Fig. 4d, e). Of note, butyrate did not affect CD4$^+$ T cell expression of HIF2α (Supplementary Fig. 10a), which is structurally similar to HIF1α[42]. Butyrate also increased HIF1α (Fig. 4f) and AhR activity (Fig. 4g). To investigate whether butyrate induction of IL-22 is dependent on the increased AhR and/or HIF1α, we treated CD4$^+$ T cells with the AhR inhibitor, CH-223191, and the HIF1α inhibitor, YC-1 or FM19G11, along with butyrate. Addition of CH-223191, YC-1, or FM19G11 decreased IL-22 expression induced by butyrate (Fig. 4h–j and Supplementary Fig. 10b). Addition of both CH-223191 and YC-1 together further suppressed the butyrate-induced IL-22 (Fig. 4h–j), suggesting AhR and HIF1α act in a synergistic manner to promote IL-22 production. Furthermore, treatment with DMOG, a stabilizer of HIF-1α, promoted IL-22 production in CD4$^+$ T cells (Supplementary Fig. 10b). However, HIF2α inhibitor, TC-S 7009, did not affect butyrate-induced IL-22 production (Supplementary Fig. 10c). All the inhibitors and agonists used did not affect cell viability (Supplementary Fig. 10d). To further confirm the role of HIF1α in butyrate-induced IL-22 production, we cultured HIF1α-deficient CD4$^+$ T cells from *Cd4*$^{cre}$*Hif1a*$^{fl/fl}$ mice with or without butyrate under Th1 conditions. IL-22 production induced by butyrate was decreased in HIF1α-deficient CD4$^+$ T cells compared with WT CD4$^+$ T cells at both mRNA and protein levels (Fig. 4k and Supplementary Fig. 11a, b). Similar results were obtained in CD4$^+$ T cells cultured under Th17 conditions, in that butyrate increased *Ahr* and *Hif1a* expression (Supplementary Fig. 12a, b), blockade of AhR and HIF1α pathways suppressed IL-22 induced by butyrate (Supplementary Fig. 12c, d), and butyrate induction of IL-22 was compromised in HIF1α-deficient CD4$^+$ T cells (Supplementary Fig. 12e). Furthermore, CD4$^+$ T cells produced higher levels of IL-22 when cultured under hypoxic conditions (3% O$_2$) than that in normoxic conditions, and butyrate treatment further increased IL-22 production in CD4$^+$ T cells under hypoxic conditions (Supplementary Fig. 10e, f). Similarly, butyrate enhanced HIF1α

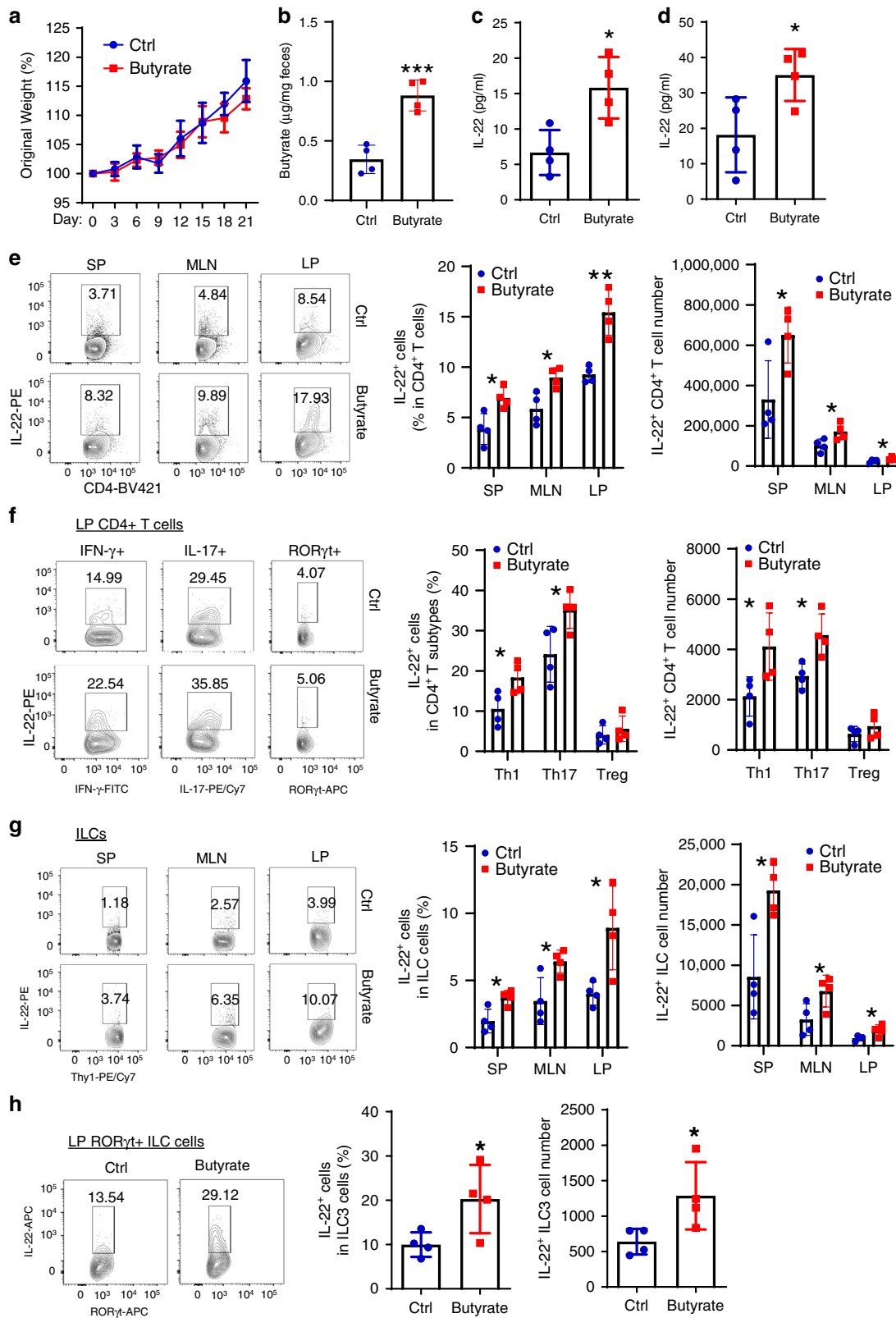

expression in ILCs (Supplementary Fig. 9d), and inhibition of HIF1α or AhR suppressed the butyrate-induced IL-22 production in ILCs (Supplementary Fig. 9e).

Next, we investigated whether butyrate induced HIF1α and AhR through GPR41 or HDAC inhibition. Treatment with GPR41 agonist, but not HDAC inhibitor, upregulated *Hif1a* and *Ahr* expression in CD4$^+$ T cells (Fig. 4l, m), indicating GPR41

mediates butyrate induction of HIF1α and AhR expression in CD4$^+$ T cells.

We have previously shown that Blimp1-mediated SCFA-induced IL-10 production in CD4$^+$ T cells[20,23]. We then explored whether Blimp1 also regulates butyrate-induced IL-22 production in CD4$^+$ T cells. Butyrate increased *Il22* expression in Blimp1-deficient CD4$^+$ T cells from *Cd4*$^{cre}$*Prdm1*$^{fl/fl}$ mice to the levels

**Fig. 2 Butyrate promotes intestinal CD4+ T cell and ILC production of IL-22.** WT mice were treated with or without 200 mM butyrate in drinking water for 3 weeks ($n = 4$ mice/group). **a** Mice were weighed daily. **b** Fecal pellets were collected prior and after 3-week treatment of butyrate, and butyrate levels were measured by LC–MS. Mice were killed on day 21, and IL-22 production in serum (**c**) and colonic organ cultures (**d**) were measured by ELISA. IL-22 production in CD4+ T cells (**e**) and ILCs (**g**) were analyzed in the spleen, MLN, and intestinal LP by flow cytometry. IL-22 levels in Th1, Th17, Treg cells (**f**), and ILCs (**h**) were measured in intestinal LP by flow cytometry. One representative of three independent experiments was shown. Data were expressed as mean ± SD. Statistical significance was tested by two-tailed unpaired Student $t$-test. **b** ***$p = 0.0009$; **c** *$p = 0.0145$; **d** *$p = 0.0393$; **e** middle panel: *$p = 0.0158$ (SP), 0.0151 (MLN), and 0.0022 (LP); right panel: *$p = 0.0359$ (SP), 0.0377 (MLN), and 0.0481 (LP); **f** middle panel: *$p = 0.0356$ (Th1) and 0.0375 (Th17); right panel: *$p = 0.0432$ (Th1) and 0.0158 (Th17); **g** middle panel: *$p = 0.0149$ (SP), 0.0227 (MLN), and 0.0232 (LP); right panel: *$p = 0.0126$ (SP), 0.0448 (MLN), and 0.0462 (LP); **h** middle panel: *$p = 0.0458$; right panel: *$p = 0.0436$.

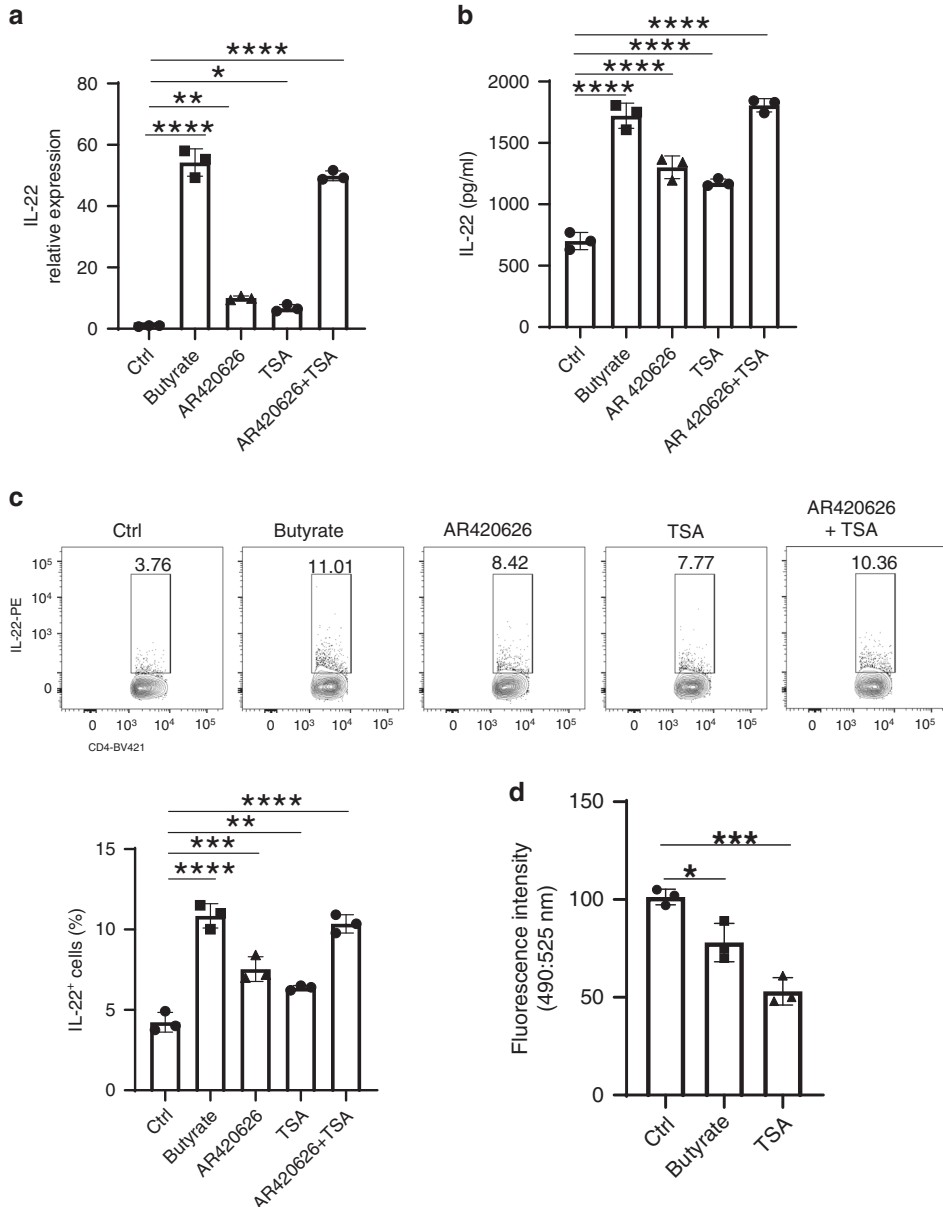

**Fig. 3 Butyrate promotes IL-22 production through GPR41 and HDAC inhibition. a**, **b** CBir1 Tg CD4+ T cells were cultured with APCs and Cbir1 peptide with or without butyrate (0.5 mM) ± AR420626 (5 μM) or/and TSA (10 mM) under Th1 conditions ($n = 3$/group). IL-22 mRNA (**a**) and protein (**b**) were measured by qRT-PCR and ELISA at 60 h. IL-22 production was measured by flow cytometry on day 5 (**c**). **d** CD4+ T cells were cultured with anti-CD3/CD28 mAbs under Th1 conditions with or without butyrate (0.5 mM) or TSA (10 mM) ($n = 3$/group). Cells were collected at 24 h for analysis of HDAC activity at fluorescence intensity at excitation/emission (490/525 nm) by using the HDAC Activity Assay Kit. One representative of three independent experiments was shown. Data were expressed as mean ± SD. Statistical significance was tested by two-tailed one-way ANOVA. **a** ****$p < 0.0001$, **$p = 0.0016$, *$p = 0.0282$; **b** ****$p < 0.0001$; **c** ****$p < 0.0001$, ***$p = 0.0002$, **$p = 0.0054$; **d** *$p = 0.0144$, ***$p = 0.0004$.

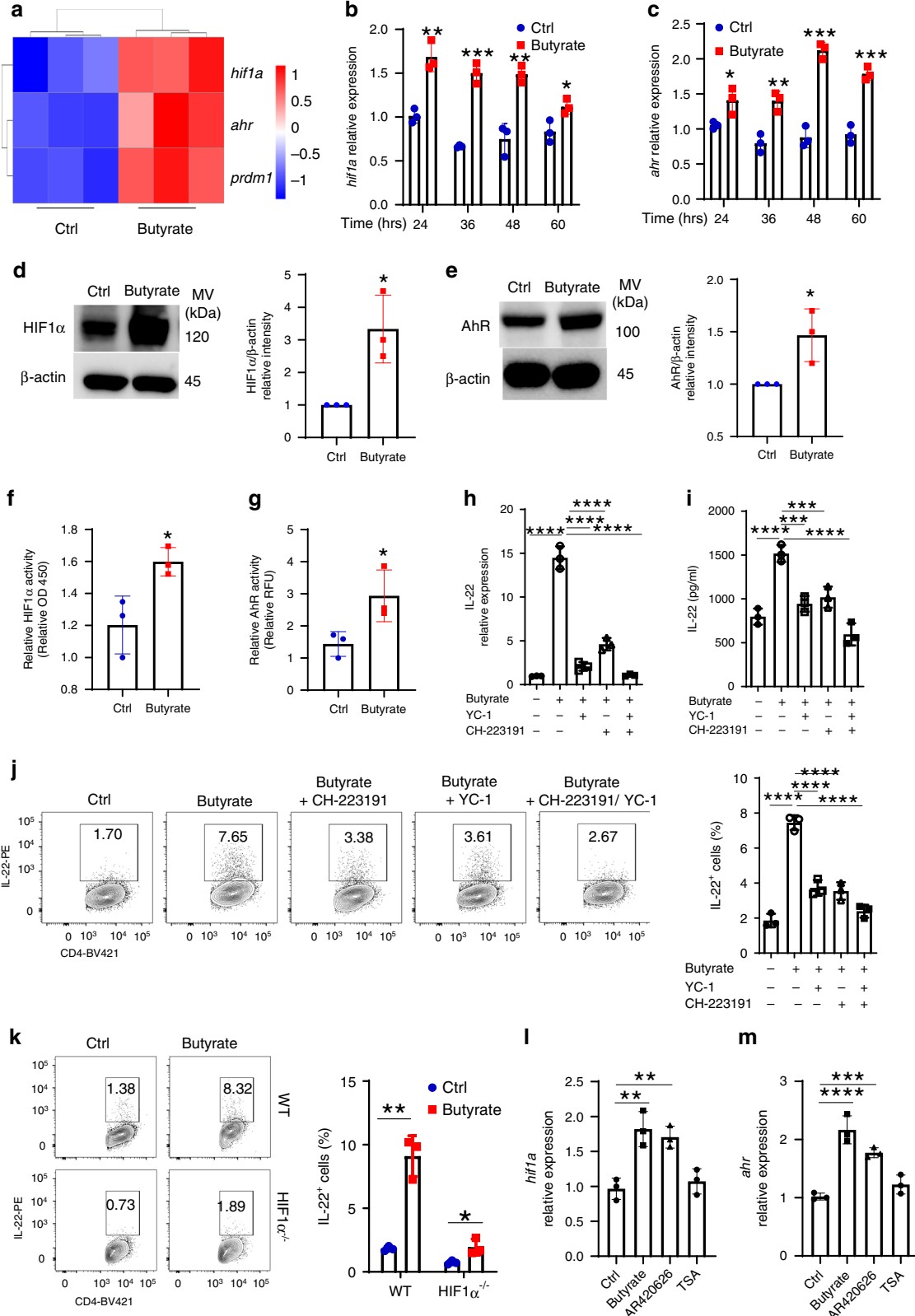

similar to WT CD4+ T cells under both Th1 and Th17 conditions (Supplementary Figs. 11c, d and 12f), suggesting that Blimp1 does not mediate butyrate induction of IL-22 in CD4+ T cells. Taken together, these results indicated that butyrate promotes IL-10 and IL-22 production in CD4+ T cells through different mechanisms.

**Stat3 and mTOR are involved in butyrate induction of IL-22.** Stat3 and mTOR have been implicated in the regulation of HIF1α/AhR expression[43–46], and SCFAs activated Stat3 and mTOR[20,21]. To explore whether butyrate increases HIF1α and AhR expression in CD4+ T cells through activation of Stat3 and mTOR, we first analyzed phosphorylation levels of Stat3 and

**Fig. 4 HIF1α and AhR mediate butyrate induction of IL-22 in CD4$^+$ T cells. a** WT CD4$^+$ T cells were activated with anti-CD3/CD28 mAbs under Th1 conditions ± butyrate (0.5 mM) for 2 days ($n = 3$ biologically independent samples per group). RNA sequencing was performed. *Hif1α*, *Ahr*, and *Prdm1* were shown in heatmap. **b–f** CD4$^+$ T cells were activated with anti-CD3/CD28 mAbs ± butyrate (0.5 mM) under Th1 conditions ($n = 3$/group). *Hif1a* (**b**) and *Ahr* (**c**) were analyzed by qRT-PCR. HIF1α (**d**) and AhR (**e**) protein was analyzed by western blot on day 2. HIF1α activity was measured using HIF1α Transcription Factor Assay Kit (**f**). **g** Raw 264.7 cells were transduced with XRE/AhR Luciferase Reporter Gene Lentivirus, and treated ± butyrate (0.5 mM) 3 days post transduction. AhR activity was assessed by luciferase. **h–j** Cbir1 Tg CD4$^+$ T cells were activated with APCs and Cbir1 peptide under Th1 conditions with butyrate (0.5 mM) ± YC-1 (5 μM) or/and CH-223191 (3 μM) for 60 h ($n = 3$/group). IL-22 mRNA (**h**) and protein (**i**) were measured by qRT-PCR and ELISA. **j** IL-22 was measured by flow cytometry on day 5. **k** WT and HIF1α$^{-/-}$ CD4$^+$ T cells were activated with anti-CD3/CD28 mAbs ± butyrate (0.5 mM) for 5 days ($n = 3$/group). IL-22 was assessed by flow cytometry. **l, m** CD4$^+$ T cells were activated with anti-CD3/CD28 mAbs under Th1 conditions with or without butyrate (0.5 mM), AR420626 (5 μM), or TSA (10 nM) for 60 h ($n = 3$/group). *Hif1a* (**l**) and *Ahr* (**m**) were measured by qRT-PCR. One representative of three independent experiments was shown (**b–m**). Data were expressed as mean ± SD. Statistical significance was tested by two-tailed unpaired Student *t*-test (**b–g**) or two-tailed one-way ANOVA (**h–m**). **b** **\*\****p = 0.0033* (24 h), *\*\*\*p = 0.0002* (36 h), *\*\*p = 0.0032* (48 h), *\*p = 0.0310* (60 h); **c** *\*p = 0.0338* (24 h), *\*\*p = 0.0054* (36 h), *\*\*\*p = 0.0003* (48 h), *\*\*\*p = 0.0007* (60 h); **d** *\*p = 0.0178*; **e** *\*p = 0.0325*; **f** *\*p = 0.0273*; **g** *\*p = 0.0435*; **h** *\*\*\*\*p < 0.0001*; **i** *\*\*\*\*p < 0.0001*, *\*\*\*p = 0.0002* (butyrate + YC-1 vs butyrate), *\*\*\*p = 0.0006*; **j** *\*\*\*\*p < 0.0001*; **k** *\*\*p = 0.0015*, *\*p = 0.0325*; **l** *\*\*p = 0.0014* (butyrate vs control) and 0.0036 (AR429626 vs control); **m** *\*\*\*\*p < 0.0001*, *\*\*\*p = 0.0009*.

mTOR in CD4$^+$ T cells treated with butyrate at different time points. Butyrate enhanced Stat3 activation at 6 h (Fig. 5a, b), whereas phosphorylated mTOR was increased 24 h after butyrate treatment (Fig. 5c, d). mTOR activation was further confirmed by increased phosphorylated S6 ribosomal protein levels (Fig. 5e), a downstream target of mTOR. To investigate the role of Stat3 and mTOR in mediating butyrate induction of IL-22, we utilized the Stat3 inhibitor, HJC0152[47], and the mTOR inhibitor, rapamycin, in CD4$^+$ T cell cultures. HJC0152 and rapamycin suppressed IL-22 mRNA and protein levels induced by butyrate in CD4$^+$ T cells under Th1 conditions (Fig. 5f, g). Interestingly, while inhibition of Stat3 decreased both *Hif1a* and *Ahr* expression induced by butyrate, mTOR inhibitor only downregulated butyrate-induced expression of *Ahr* but not *Hif1a* (Fig. 5h, i), suggesting Stat3 and mTOR activation differentially regulates HIF1α and AhR to promote IL-22 production in CD4$^+$ T cells. Furthermore, while HJC0152 did not affect mTOR activation (Supplementary Fig. 13a), rapamycin moderately suppressed phosphorylated Stat3 induction by butyrate (Supplementary Fig. 13b) in CD4$^+$ T cells. Addition of mTOR inhibitor further reduced butyrate-induced IL-22 production suppressed by Stat3 inhibitor in CD4$^+$ T cells (Supplementary Fig. 13c). The role of Stat3 in butyrate-induced IL-22 production under Th1 conditions was further confirmed in Stat3-deficient CD4$^+$ T cells from *Cd4*$^{Cre}$*Stat3*$^{fl/fl}$ mice, in that butyrate-induced IL-22 production was decreased in Stat3-deficient CD4$^+$ T cells compared to WT CD4$^+$ T cells (Fig. 5j and Supplementary Fig. 11e, f). In addition, we obtained similar results for the roles of Stat3 and mTOR in butyrate-induced IL-22 production and expression of HIF1α and AhR in CD4$^+$ T cells cultured under Th17 conditions (Supplementary Fig. 12g–l). Similarly, butyrate activated Stat3 and mTOR (Supplementary Fig. 9f, g), and inhibition of Stat3 and mTOR suppressed their ability to produce IL-22 in ILCs (Supplementary Fig. 9h).

**Butyrate promotes HIF1α binding to the *Il22* promoter.** Upon hypoxia, HIF1α dimerizes with HIF1β, together with co-activators, to translocate into the nucleus to regulate target gene expression by binding to the hypoxia response element (HRE), NCGTG, in the promoters of the target genes. By retrieving genomics data in PubMed and Ensembl and using the consensus core (NCGTG), we found the putative HRE on *Il22* promoter region (−2000 bp) (Fig. 6a). To confirm the direct binding of HIF1α to the *Il22* promoter in CD4$^+$ T cells, we performed a CHIP assay in CD4$^+$ T cells under Th1 conditions. Compared to the control with anti-IgG antibody, qRT-PCR amplification of DNA that was immunoprecipitated with anti-HIF1α antibody resulted in specific enrichment of the HRE region in the *Il22* promoter (Fig. 6b), suggesting HIF1α directly binds to the *Il22*

promoter. Next, we asked whether butyrate enhances the binding of HIF1α to the *Il22* promoter. We treated CD4$^+$ T cells with or without butyrate under Th1 conditions for 48 h. Butyrate enhanced HIF1α binding to the HRE region of the *Il22* promoter (Fig. 6c).

**Butyrate induces histone acetylation of HRE on *Il22* promoter.** Histone modifications, such as acetylation and methylation, are associated with gene transcription through regulation of accessibility to the target DNA. Lysine 9 (K9) acetylation on histone H3 (H3K9ac) and K4 trimethylation on histone H3 (H3K4me3), which are co-localized in gene promoters, are the active and repressive markers of transcription, respectively[48]. Given that butyrate is an effective HDAC inhibitor, we investigated whether butyrate suppresses HDAC to facilitate HIF1α binding to the *Il22* promoter. We performed a CHIP assay using antibodies against H3K9ac and H3K9me3 to determine the accessibility of HIF1α-binding sites in the *Il22* promoter in CD4$^+$ T cells cultured with or without butyrate under Th1 conditions. Butyrate increased the H3K9 acetylation, but suppressed the trimethylation of H3K9 in HRE sites of the *Il22* promoter (Fig. 6d, e), indicating butyrate increases the accessibility of HIF1α-binding sites in the *Il22* promoter through histone modification.

**Butyrate inhibits colitis through promoting IL-22.** IL-22 is a crucial cytokine in host defense against enteric infection of *Citrobacter rodentium* (*C. rodentium*), which is similar to human enteropathogenic *Escherichia coli* (EPEC) associated with IBD[49]. We then investigated whether butyrate protects intestines from *C. rodentium* infection and the role of IL-22 invovled. We infected WT mice orally with *C. rodentium* ($5 × 10^8$ CFU/mice) on day 0 and administrated with or without butyrate in drinking water for 10 days. The mice treated with butyrate exhibited less weight loss (Fig. 7a), and decreased intestinal inflammation (Fig. 7b), with higher IL-22 and IL-10 production in intestinal LP CD4$^+$ T cells (Fig. 7c). IFN-γ$^+$ CD4$^+$ T cells, but not IL-17$^+$CD4$^+$ T cells, were decreased in the intestinal LP in mice treated with butyrate (Fig. 7c). In addition, butyrate supplementation increased IL-22 production, but did not affect IL-17 production, in the intestinal ILCs (Fig. 7d). Administration of butyrate decreased the fecal *C. rodentium* CFU (Fig. 7e) and *C. rodentium* CFU in liver (Fig. 7f), indicating butyrate promotes *C. rodentium* clearance in the colon and decreases *C. rodentium* dissemination to the liver.

To determine whether ILCs, CD4$^+$ T cells, or both ILCs and CD4$^+$ T cells are required for the effects of butyrate on intestinal inflammation, we infected mice with *C. rodentium* ($5 × 10^8$ CFU/mice) orally on day 0 followed by treatment with or without

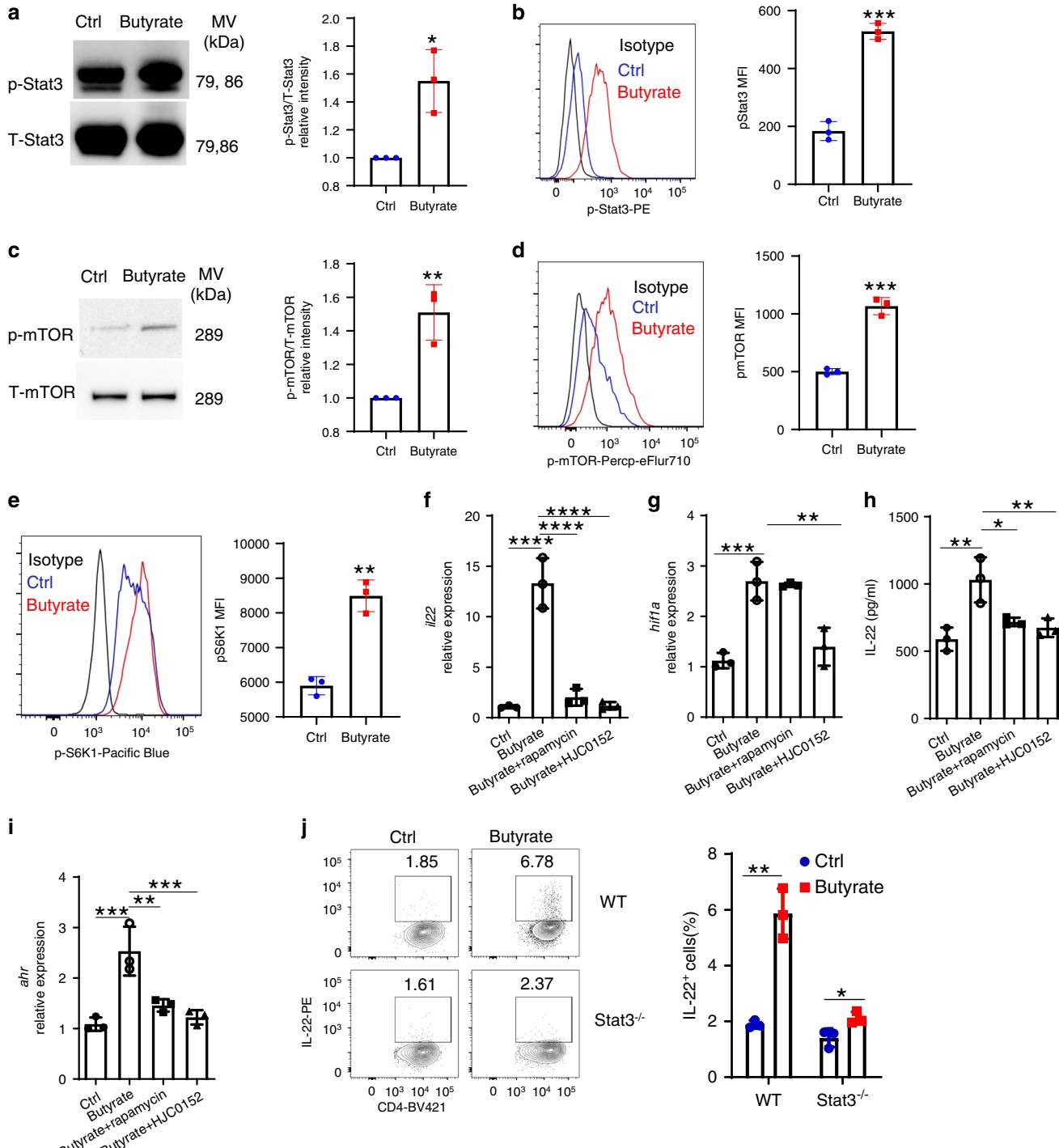

**Fig. 5 Stat3 and mTOR regulate IL-22 production by CD4$^+$ T cells. a–d** WT CD4$^+$ T cells were activated with anti-CD3/CD28 mAbs under Th1 conditions with or without butyrate (0.5 mM) ($n = 3$/group). Phosphorylated Stat3 (6 h) (**a, b**) and phosphorylated mTOR (24 h) (**c, d**) were assessed by western blot and flow cytometry. Phosphorylated S6K was analyzed by flow cytometry (**e**). **f–i** CBir1 Tg CD4$^+$ T cells were activated with APCs and CBir1 peptide under Th1 conditions with butyrate (0.5 mM) ± rapamycin (1 μM) or HJC0152 (1 μM). IL-22 mRNA (**f**) and protein (**g**) were assessed by qRT-PCR and ELISA at 60 h ($n = 3$/group). Expression of *Hif1a* (**h**) and *Ahr* (**i**) was analyzed at 48 h by qRT-PCR. **j** WT and Stat3$^{-/-}$ CD4$^+$ T cells were treated with or without butyrate (0.5 mM) for 5 days ($n = 3$/group). IL-22 production was measured by flow cytometry. One representative of three independent experiments was shown. Data were expressed as mean ± SD. Statistical significance was tested by two-tailed unpaired Student *t*-test (**a–e, j**) or two-tailed one-way ANOVA (**f–i**). **a** *$*p = 0.0134$; **b** ***$p = 0.0002$; **c** **$p = 0.0059$; **d** ***$p = 0.0002$; **e** **$p = 0.0010$; **f**, ****$p < 0.0001$; **g** **$p = 0.0019$ (butyrate vs control) and 0.0069 (butyrate + rapamycin vs butyrate), *$p = 0.0141$; **h** ***$p = 0.0004$, **$p = 0.0012$; **i**, ***$p = 0.0004$ (butyrate vs control) and 0.009 (butyrate + HJC0152 vs butyrate), **$p = 0.0030$; **j** **$p = 0.0017$, *$p = 0.0338$.

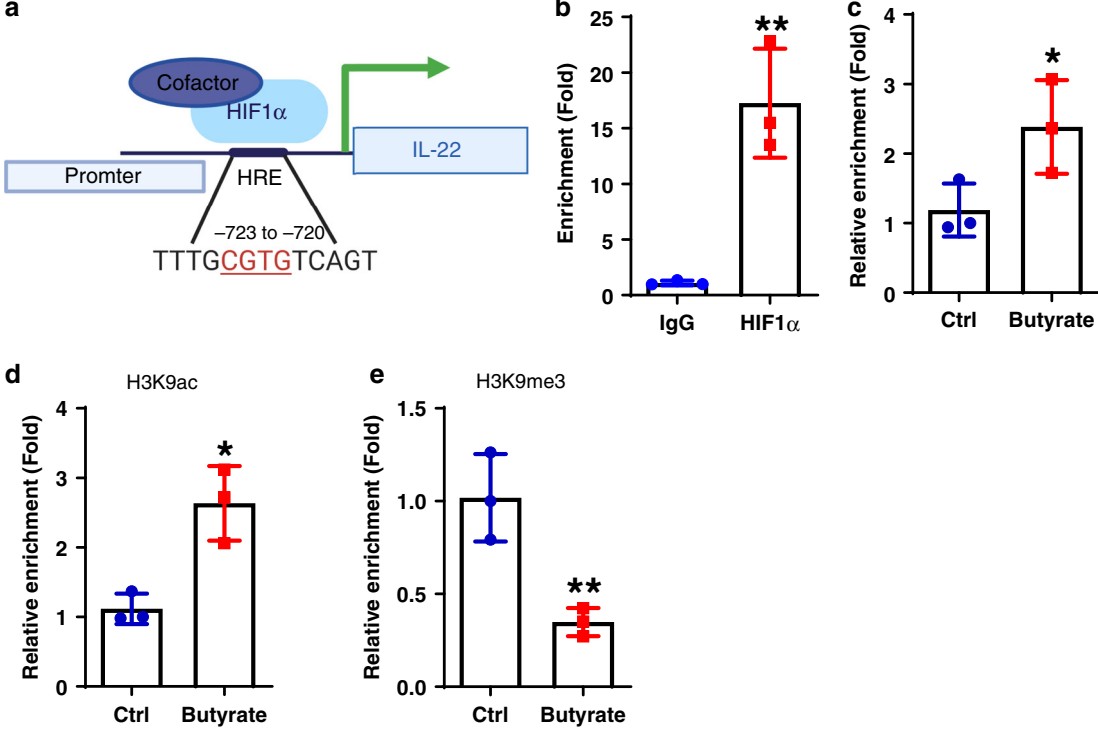

**Fig. 6 Butyrate promotes HIF1α binding to Il22 promoter in CD4+ T cells. a** Schematic diagram of HIF1α binding to *Il22* promoter. **b** WT CD4+ T cells were activated with anti-CD3/CD28 mAbs under Th1 conditions for 2 days (*n* = 3/group). HIF1α binding to *Il22* promoter was analyzed by CHIP assay. **c–e** WT CD4+ T cells were cultured under Th1 conditions with or without butyrate (0.5 mM) for 2 days (*n* = 3/group). HIF1α binding to *Il22* promoter was analyzed by CHIP assay (**c**). The H3K9 acetylation (**d**) and trimethylation (**e**) levels in HIF1α-binding site on *Il22* promoter were assessed by CHIP assay. One representative of three independent experiments (**b**, **c**), or two independent experiments (**d**, **e**) was shown. Data were expressed as mean ± SD. Statistical significance was tested by two-tailed unpaired Student *t*-test. **b** **$p$ = 0.0047; **c** *$p$ = 0.0278; **d** *$p$ = 0.0105; **e** **$p$ = 0.0094.

butyrate in drinking water for 10 days. Groups of the mice received either anti-CD4 mAb to deplete CD4+ T cells, anti-Thy1 mAb to deplete CD4+ T cells and ILCs, or control IgG. Depletion of CD4+ T cells led to more severe colitis, and depletion of both CD4+ T cells and ILCs further aggravated the severity of colitis (Fig. 7g), indicating that both CD4+ T cells and ILCs are critical in protecting the intestines against *C. rodentium* infection. Feeding butyrate provided partial protection against *C. rodentium* infection in CD4+ T cell-depleted mice (Fig. 7g). However, depletion of both CD4+ T cells and ILCs completely abrogated the protective effects of butyrate (Fig. 7g). Consistent with the severity of the disease, administration of butyrate decreased fecal CFU in CD4+ T cell-depleted mice to a lesser degree than in control mice, but was unable to affect fecal CFU in mice depleted of both CD4+ T cells and ILCs (Fig. 7h). This indicates that both ILCs and CD4+ T cells are important in butyrate protection of the intestines from *C. rodentium* infection.

To investigate whether increased IL-22 mediates butyrate protection against *C. rodentium* infection, we infected WT and *Il22*−/− mice orally with *C. rodentium* and treated with or without butyrate in drinking water. *Il22*−/− mice suffered more weight loss compared with WT mice, and butyrate administration decreased weight loss in WT mice, but not in *Il22*−/− mice (Fig. 8a). *Il22*−/− mice showed more severe colitis compared with WT control mice. Feeding butyrate decreased colitis severity in WT but not in *Il22*−/− mice (Fig. 8b). In addition, butyrate treatment decreased colonic IL-6 and TNF levels in WT mice but not in *Il22*−/− mice (Fig. 8c, d). Fecal *C. rodentium* CFU was decreased in WT but not *Il22*−/− mice treated with butyrate (Fig. 8e), indicating that butyrate promotes *C. rodentium* clearance in the gut in an IL-22-dependent manner. We also

checked the *C. rodentium* dissemination in the liver. *C. rodentium* CFU in liver was significantly higher in *Il22*−/− mice compared with WT mice. Butyrate treatment decreased CFU levels in the liver of WT mice but not *Il22*−/− mice (Fig. 8f), suggesting butyrate limits *C. rodentium* dissemination from the intestine to the liver through upregulating IL-22 production.

To investigate whether butyrate-induced IL-22 regulates intestinal inflammation upon inflammatory insult, we assessed the role of IL-22 in butyrate inhibition of Dextran sulfate sodium (DSS)-induced colitis. Similar to the *C. rodentium* model, *Il22*−/− control mice showed more weight loss and developed more severe colitis compared with WT control mice (Supplementary Fig. 14a, b). Butyrate-treated WT mice showed less weight loss compared with control WT mice, while there was no difference in weight loss between butyrate-treated IL-22−/− mice and control *Il22*−/− mice (Supplementary Fig. 14a). Administration of butyrate alleviated intestinal inflammation in WT but not in *Il22*−/− mice, characterized by less inflammation and lower histopathological scores (Supplementary Fig. 14b), and decreased levels of IL-6 and TNFα in colonic organ culture (Supplementary Fig. 14c, d). In addition, butyrate increased IL-22+CD4+ T cells, but did not affect the percentage of IFN-γ+ CD4+ T cells and IL-17+CD4+ T cells, in the intestinal LP of DSS-treated WT mice (Supplementary Fig. 14e). Consistent with our previous study, butyrate promoted IL-10 production in LP CD4+ T cells of WT mice (Supplementary Fig. 14e). Furthermore, butyrate promoted IL-22, but not IL-17, production in LP ILCs (Supplementary Fig. 14f). Taken together, these results demonstrated that butyrate protects the intestines from inflammation induced by both enteric infection and intestinal injury through the upregulation of IL-22 production.

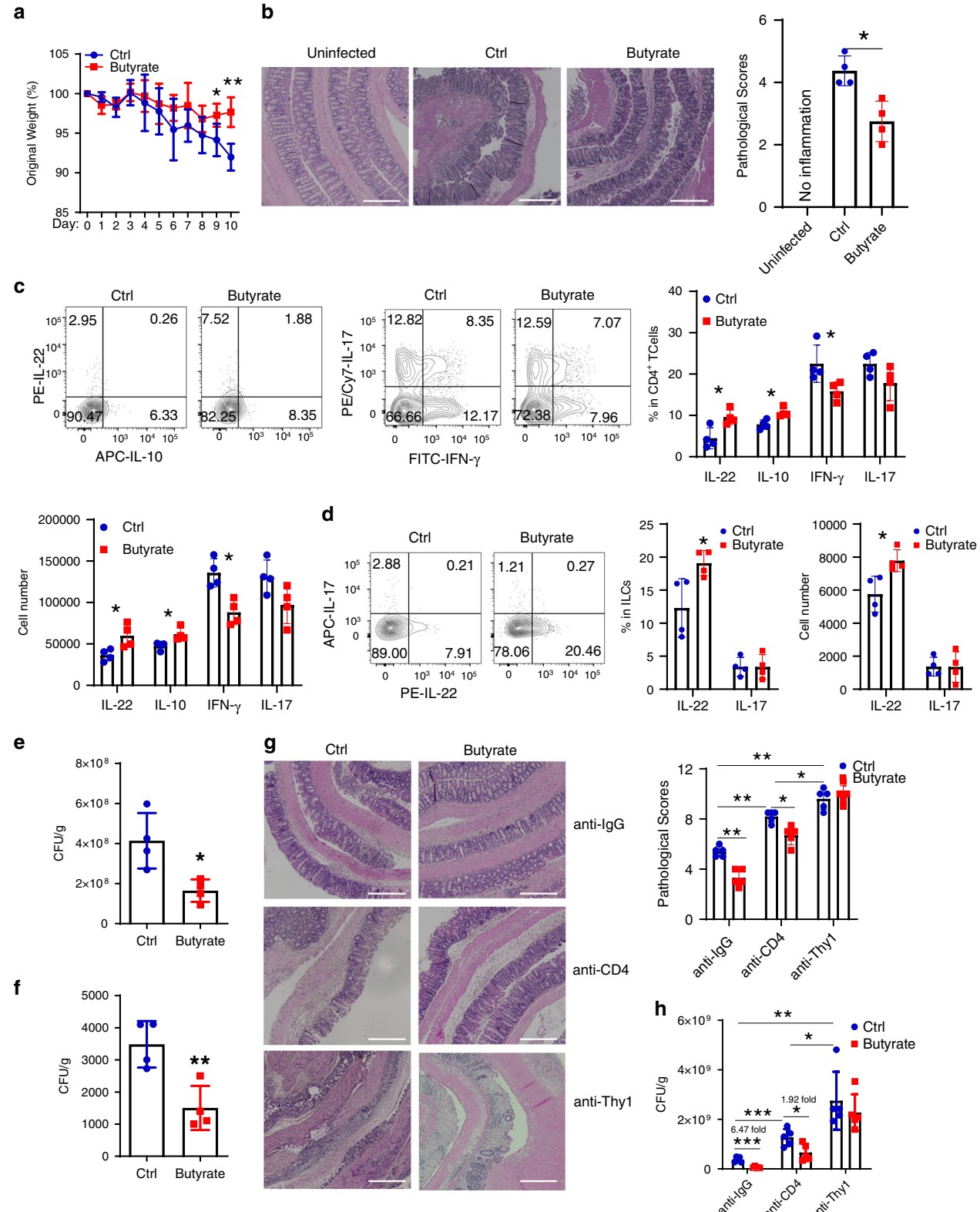

**Butyrate promotes human CD4+ T cell IL-22 production**. To assess whether butyrate induces CD4+ T cell IL-22 production in IBD patients for translational potential, we treated peripheral blood CD4+ T cells, isolated from healthy volunteers, and patients with active Crohn's disease (CD) and ulcerative colitis (UC), with anti-CD3 mAb and anti-CD28 mAb with or without butyrate. Butyrate promoted *Il22* mRNA levels in human CD4+

T cells, including healthy volunteers, CD, and UC patients (Fig. 9a). The percentage of IL-22+CD4+ T cells and IL-22 production were increased after treatment with butyrate (Fig. 9b, c). Similar to the results from the mouse study, butyrate increased *Hif1a* and *Ahr* expression in human CD4+ T cells (Fig. 9d, e). Furthermore, inhibition of HIF1α and AhR using HIF1α inhibitor YC-1 and AhR inhibitor CH-223191 suppressed butyrate-induced

**Fig. 7 Butyrate protects the intestines from *Citrobacter rodentium* infection. a–f** WT mice ($n = 4$ mice/group) were orally infected with *Citrobacter rodentium* (*C. rodentium*, $5 \times 10^8$ CFU/mice), and treated with or without butyrate (200 mM) in drinking water for 10 days. Mice were weighed daily (**a**), and killed on day 10. Colonic histopathology (**b**), LP IL-22+, IL-10+, IFN-γ+, and IL-17+ CD4+ T cells (**c**), and IL-22+ and IL-17+ ILCs (**d**) were measured. CFU in feces (**e**) and liver (**f**) were measured. **g, h** WT mice ($n = 4$ mice/group) were orally infected with *C. rodentium* ($5 \times 10^8$ CFU/mice) on day 0, and with or without butyrate (200 mM) in drinking water for 10 days. Mice were administered with anti-IgG antibody (25 mg/kg), anti-CD4 antibody (25 mg/kg), or anti-Thy1 (25 mg/kg) i.p. every other day. Mice were killed on day 10. Colonic histopathology was assessed (**g**), and CFU in feces was measured (**h**). One representative of three independent experiments (**a–f**) or two independent experiments (**g, h**) was shown. Scale bar, 300 μm (**b, g**). Data were expressed as mean ± SD. Statistical significance was tested by two-tailed unpaired Student *t*-test (**a, c–f, h**), the non-parametric two-tailed Mann–Whitney *U* test (**b, g**). **a** *$p = 0.0477$, **$p = 0.0041$; **b** *$p = 0.0286$; **c** right: *$p = 0.0169$ (IL-22), 0.0107 (IL-10), and 0.0387 (IFN-γ); lower left: *$p = 0.0247$ (IL-22), 0.0224 (IL-10), and 0.0060 (IFN-γ); **d** middle: *$p = 0.0308$; right: *$p = 0.0188$; **e** *$p = 0.0157$; **f** **$p = 0.0073$; **g** **$p = 0.0079$, *$p = 0.0159$ (anti-CD4 + butyrate vs anti-CD4) and 0.0317 (anti-CD4 vs anti-Thy1); **h**, ***$p = 0.0002$ (anti-IgG + butyrate vs anti-IgG) and 0.0005 (anti-CD4 vs anti-IgG), *$p = 0.0214$ (anti-CD4 + butyrate vs anti-CD4) and 0.0263 (anti-Thy1 vs anti-CD4), **$p = 0.0020$.

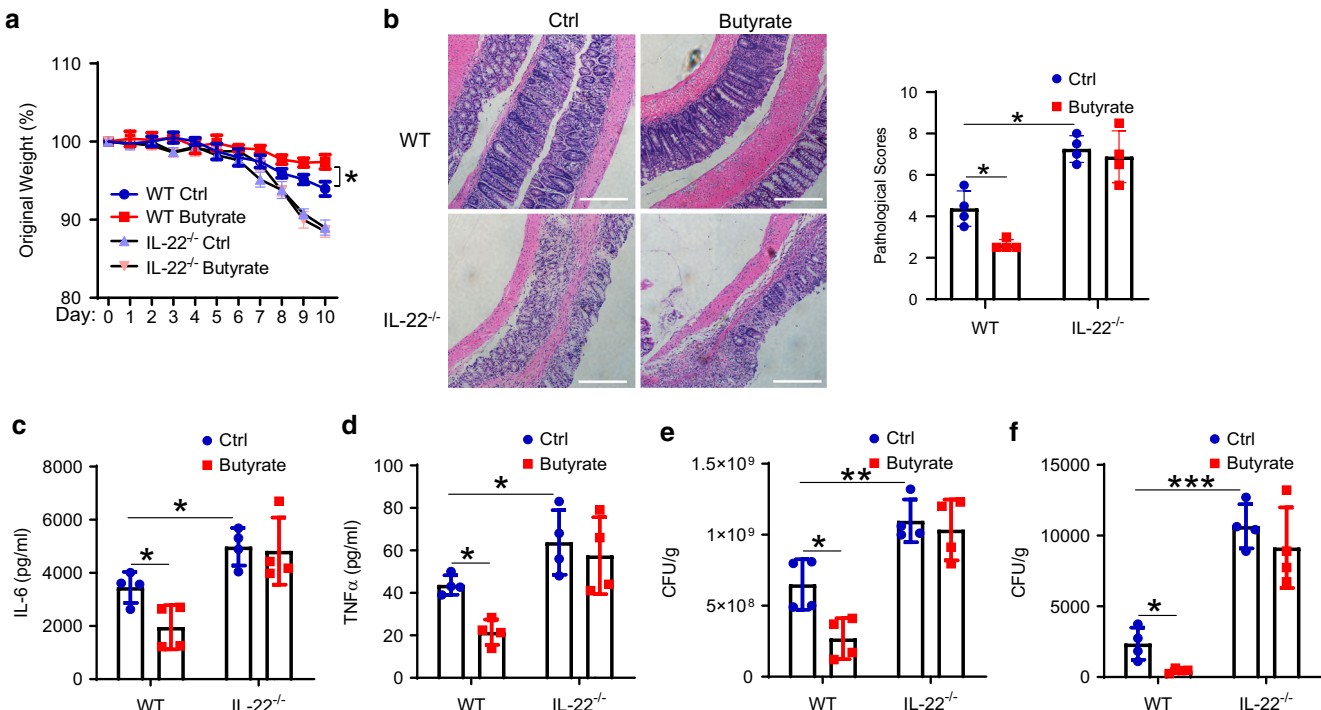

**Fig. 8 Butyrate inhibits intestinal infection by promoting IL-22 production.** WT and IL-22$^{-/-}$ mice ($n = 4$ mice/group) were orally infected with *C. rodentium* ($5 \times 10^8$ CFU/mice) on day 0, and treated with or without butyrate (200 mM) in drinking water for 10 days. Mice were weighed daily (**a**). At day 10, colonic histopathology was analyzed (**b**), and colonic IL-6 (**c**) and TNF (**d**) production in colonic tissue was determined by ELISA. CFU in feces (**e**) and liver (**f**) were measured. One representative of three independent experiments was shown. Scale bar, 300 μm (**b**). Data were expressed as mean ± SD. Statistical significance was tested by two-tailed unpaired Student *t*-test (**a, c–f**) or the nonparametric two-tailed Mann–Whitney U test (**b**). **a** *$p = 0.0379$; **b** *$p = 0.0286$; **c** *$p = 0.0254$ (WT butyrate vs WT control) and 0.0156 (IL-22$^{-/-}$ control vs WT control); **d** *$p = 0.00105$ (WT butyrate vs WT control) and 0.0452 (IL-22$^{-/-}$ control vs WT control); **e** *$p = 0.0155$, **$p = 0.0086$; **f**, *$p = 0.0163$, ***$p = 0.0001$.

IL-22 production (Fig. 9f), indicating that butyrate promotes IL-22 production in human CD4+ T cells, including IBD patients, through regulation of HIF1α and AhR.

## Discussion

Emerging evidence indicates that interaction between microbiota and IL-22 is central at barrier sites in the regulation of intestinal homeostasis. IL-22 regulates the gut microbiota composition through promoting intestinal barrier function by inducing epithelial cell production of antimicrobial peptides, mucins, and other beneficial effects. On the other hand, gut microbiota also regulates intestinal IL-22 production, in which the mechanisms are still not well-established. Many functions of gut microbiota in modulating health and disease are through their metabolites[50,51]. Our current study demonstrated that SCFAs, the major microbiota metabolites from high fiber diet, promote CD4+ T cells and

ILC IL-22 production through upregulating AhR and HIF1α, and, thus, provided novel insights into how gut microbiota, through their metabolites SCFAs, regulates IL-22 production to maintain the intestinal homeostasis.

As IL-22 is critical in the regulation of intestinal barrier function and intestinal homeostasis, how its production is regulated has been intensively investigated. Interestingly, although both ILC3 and CD4+ T cells produce IL-22 under steady conditions, most insights into how IL-22 production is regulated in both cells are achieved either upon enteric infection or intestinal inflammation. Several proinflammatory cytokines have been identified critical in mediating IL-22 production, including IL-23, IL-6, and IL-1β[6,52,53]. As IL-22 production in the intestines depends on gut microbiota, this begs the questions on what products of gut microbiota function as inducers of IL-22 production. It has been shown that AhR ligands produced by specific gut bacteria species are able to stimulate ILC3 and CD4+ T cells

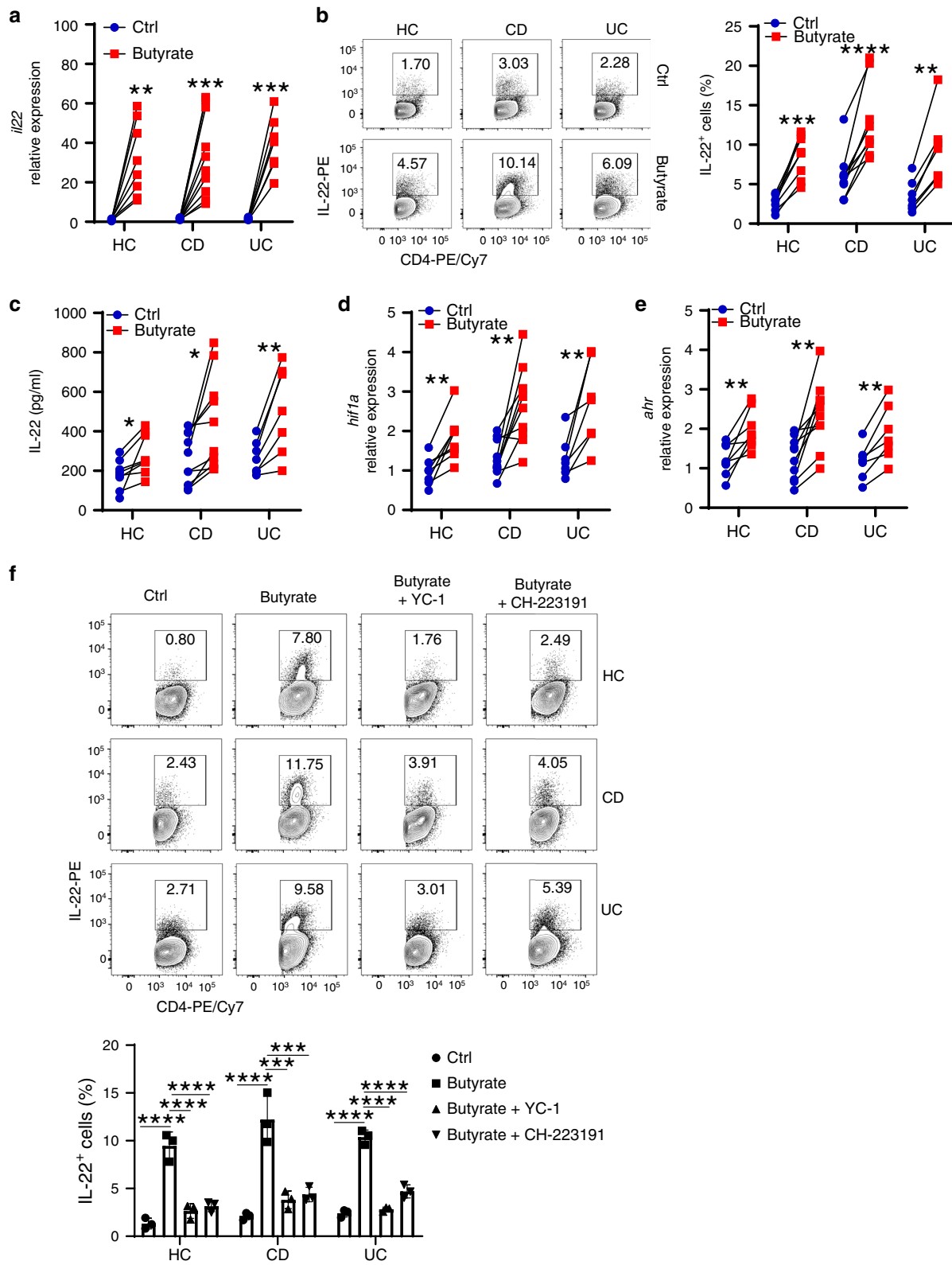

to produce IL-22[14,15]. Colonization of *Clostridia*, which produces SCFAs[18,19], in antibiotic-treated neonatal mice induces IL-22 production in ILCs and CD4[+] T cells[17], which raised the question whether *Clostridia* promote IL-22 production through upregulating SCFAs production. SCFAs as major microbiota metabolites from high fiber diets are present in the intestinal lumen at high concentrations, which make them the potential major players in the induction of IL-22 production in the intestines. Our study demonstrated that SCFAs induce IL-22 production in CD4[+] T cells and ILCs. Consistently, a recent report showed GPR43, one of the major receptors for SCFAs, promotes ILC3 development and function[27], further supporting a role of the SCFAs in promoting IL-22 production in both ILCs and CD4[+] T cells in the intestines to maintain the intestinal homeostasis.

**Fig. 9 Butyrate induces human CD4⁺ T cell IL-22 production. a–e** Peripheral blood CD4⁺ T cells were isolated from healthy controls (HC, $n = 8$ biologically independent samples), patients with active Crohn's colitis (CD, $n = 10$ biologically independent samples) and ulcerative colitis (UC, $n = 7$ biologically independent samples), and activated with anti-CD3/CD28 mAbs with or without butyrate (0.5 mM). *Il22* expression was assessed at day 3 by qRT-PCR (**a**), IL-22⁺ cells were measured by flow cytometry at day 5 (**b**), and IL-22 production in supernatants was measured at day 3 by ELISA (**c**). *Hif1a* (**d**) and *Ahr* (**e**) expression in CD4⁺ T cells were analyzed by qRT-PCR at day 3. **f** Peripheral blood CD4⁺ T cells from healthy controls, CD, and UC patients were treated with or without butyrate (0.5 mM) ± YC-1 (20 μM) or CH-223191 (5 μM) for 5 days ($n = 3$/group). IL-22 production was analyzed by flow cytometry. One representative of three independent experiments was shown. Scale bar, 300 μm. Data were expressed as mean ± SD. Statistical significance was tested by two-tailed paired Student *t*-test (**a–e**), or two-tailed one-way ANOVA (**f**). **a** **$p = 0.0024$, ***$p = 0.0008$ (CD), and 0.0003 (UC); **b** ***$p = 0.0001$, ****$p < 0.0001$, **$p < 0.0011$; **c** *$p = 0.0185$ (HC) and 0.0130 (CD), **$p = 0.0046$; **d** **$p = 0.0023$ (HC), 0.0024 (CD), and 0.0094 (UC); **e** **$p = 0.0040$ (HC), 0.0014 (CD), and 0.0023 (UC); **f** ****$p < 0.0001$, ***$p = 0.003$ (CD butyrate + YC-1 vs CD butyrate) and 0.004 (CD butyrate + CH-223191 vs CD butyrate).

---

Our previous study showed butyrate promotes differentiated Th1 cell IL-10 production via Blimp1 and GPR43 pathways. Although we confirmed butyrate increased Blimp1 expression in CD4⁺ T cells, loss of Blimp1 and GPR43 did not affect the IL-22 expression induced by butyrate. Furthermore, we found GPR41 mediates butyrate induction of IL-22 production in CD4⁺ T cells and ILCs. Thus, butyrate regulates CD4⁺ T cells production of IL-10 and IL-22 through different mechanisms. Indeed, we found that butyrate-treated CD4⁺ T cells expressed higher levels of AhR expression, a master regulator of IL-22, and inhibition of AhR suppressed butyrate-induced IL-22. HIF1α, a subunit of HIF1[54], has been shown to regulate the functions of different CD4⁺ T cells. Butyrate was reported to increase oxygen consumption to activate HIF1 in intestinal epithelial cells[55]. Interestingly, HIF1α was also increased in butyrate-treated CD4⁺ T cells and ILCs, and blockade of HIF1α both pharmacologically and genetically suppressed butyrate-induced IL-22, indicating a crucial role of HIF1α in mediating butyrate induction of IL-22. Furthermore, hypoxia promoted CD4⁺ T cell IL-22 production, and treatment with butyrate also promoted CD4⁺ T cell IL-22 expression under hypoxia condition. This result has a particular significance for SCFAs induction of IL-22 in the intestines in vivo. It has been shown that physiological hypoxia predominates in the normal intestinal mucosa, especially in the colon, and the inflammatory lesions in the inflamed intestines of the experimental colitis are profoundly hypoxic or even anoxic[56,57]. Our data that SCFAs promote CD4⁺ T cell IL-22 production under hypoxia condition suggest SCFAs as major inducers of CD4⁺ T cell IL-22 production in the intestines under both physiological and inflammatory conditions through induction HIF1α. HIF1α has been shown to promote Th17 cell IL-17 production through association with RORγt at the *Il17* promoter[58]. Although butyrate treatment increases HIF1α in Th17 cells, it inhibits their IL-17 production, which is likely due to the decreased expression of Rorα and Rorγt as we recently reported[23]. Our study, thus, identifies HIF1α as a transcription factor for IL-22 production in CD4⁺ T cells and ILCs.

HIF1α modifies several target genes by binding to the HRE in their promoters[59]. In this study, we demonstrated that HIF1α directly binds to the *Il22* promoter. More interestingly, butyrate promoted hypoxia binding to the *Il22* promoter through increasing the H3K9 acetylation and suppressing the H3K9 tri-methylation in HRE sites of IL-22 promoter. SCFAs have been shown as potent HDAC inhibitor, which modifies chromatin[37]. We found TSA, a HDAC inhibitor, upregulated IL-22 production in CD4⁺ T cells and ILCs, which mimics butyrate induction of IL-22 production, suggesting butyrate promotes IL-22 production at least partially through inhibition of HDAC.

Consistent with in vitro data, butyrate supplementation increased IL-22 production in intestinal LP CD4⁺ T cells and ILCs under both steady conditions and inflammatory conditions. Butyrate administration ameliorated *C. rodentium* infection severity, promoted *C. rodentium* clearance, and decreased *C. rodentium* dissemination from colon to liver in WT mice, but not *Il22*⁻/⁻ mice, suggesting an indispensable role of butyrate-induced IL-22 production in butyrate protective role for host defense. Consistent with the previous study[20], butyrate administration protected mice from colitis in WT mice. However, butyrate supplementation did not protect *Il22*⁻/⁻ mice from DSS-induced colitis, although butyrate treatment in vivo increased both IL-22 and IL-10 production in mice, which suggests that butyrate induction of IL-10 alone is insufficient to decrease colitis severity in this model. Butyrate can modulate oxygen availability in the intestines, and the gut microbiota can shift drastically based on oxygen availability[60,61]. Previously, we showed that butyrate affects gut microbiota composition in mice[21]. In the current study, we found that the administration of butyrate increased butyrate levels in the colon. Thus, butyrate induction of IL-22 in vivo is likely due to the combination of altered gut microbiota and increased butyrate in the intestines. Interestingly, mice administrated with butyrate showed goblet cell hyperplasia, which might be due to higher IL-22 production that has been found to mediate goblet cell hyperplasia[62].

Overall, we demonstrate gut microbiota-derived metabolites SCFAs promote CD4⁺ T cell and ILC production of IL-22 through GPR41 and HDAC inhibition. HIF1α and AhR are mediated in the butyrate induction of IL-22, which is differentially regulated by mTOR and Stat3 (Fig. 10), thus, providing a novel function of gut microbiota-derived metabolites in the regulation of intestinal homeostasis. Interestingly, SCFAs also promote human CD4⁺ T cell IL-22 production, even from IBD patients, our study provides SCFAs as a new potential therapeutic target for suppressing intestine infection and inflammation, and eventually treatment of IBD patients.

## Methods

**Mice.** Wild-type (WT) C57BL/6J mice were purchased from Jackson Laboratory. *Prdm1*fl/fl mice and *Stat3*fl/fl mice were purchased from Jackson Laboratory, and bred to B6.*Cd4*cre mice from Jackson Laboratory. *Gpr43*⁻/⁻ (*Ffar2*tm1Lex) mice were obtained from Bristol-Myers Squibb. *Il22*⁻/⁻ mice on the B6 background were obtained from Amegen (Thousand Oaks, CA). *Gpr109a*⁻/⁻ mice were obtained from Dr. Nagendra Singh of the Augusta University. *Cd4*cre*Hif1a*fl/fl mice were obtained from Dr. Fan Pan of Sidney Kimmel Comprehensive Cancer Center. CBir1 TCR transgenic (CBir1 Tg) mice were bred in the Animal Resource Center of University of the Texas Medical Branch (UTMB). All the mice were maintained on a 12 h-light/dark cycle and at the temperature of 20–26 °C with 30–70% humidity in the specific pathogen-free animal facilities. The animal care and care were in accordance with institutional guidelines of UTMB, and all experiments were approved by the Institutional Animal Care and Use Committee of UTMB.

**Human.** Patients with active Crohn's disease (CD) and Ulcerative colitis (UC) were recruited at the department of gastroenterology, Shanghai 10th People's Hospital of Tongji University (Shanghai, China). The diagnosis of CD and UC was based on clinical symptoms, endoscopic examination and histological finding, and Crohn's disease activity index (CDAI) or Mayo scores for UC were used to evaluate the disease severity. EDTA anticoagulated peripheral blood samples were collected from eight healthy volunteers, ten active CD patients, and seven active UC patients.

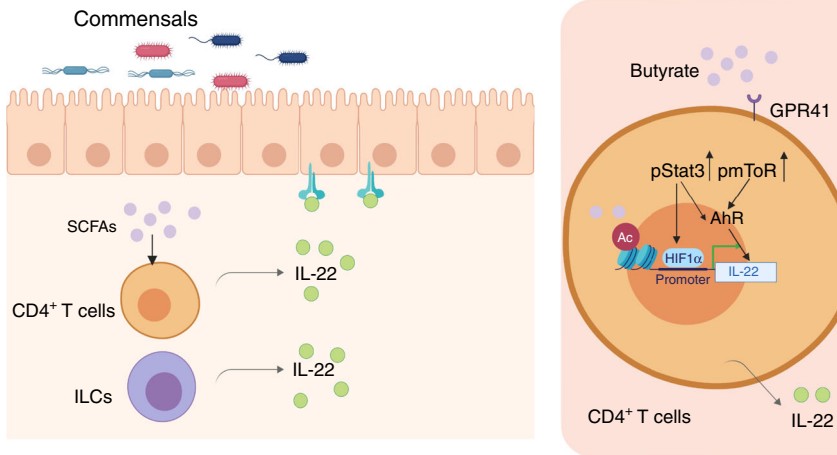

**Fig. 10 SCFAs induction of IL-22 in CD4$^+$ T cells and ILCs.** Gut microbiota-derived SCFAs promote IL-22 production in CD4$^+$ T cells and ILCs. Mechanically, butyrate promotes IL-22 production through GPR41 and HDAC inhibition. Furthermore, butyrate upregulates HIF1α and AhR, which is differentially regulated by mTOR and Stat3. HIF1α directly binds to the *Il22* promoter, and butyrate increases HIF1α binding to the *Il22* promoter through histone modifications.

Written informed consent was obtained from all participants, and all the human studies were approved by the Institutional Review Board for Clinical Research of Shanghai Tenth People's Hospital, Tongji University. The characteristics of patients and healthy volunteers are described in Supplementary Table 1.

**CD4$^+$ T cells isolation and culture.** Mouse CD4$^+$ T cells were isolated from spleen or mesenteric lymph node (MLN) using anti-mouse CD4 magnetic particles (Cat#551539, BD Biosciences). CD4$^+$ T cells were seeded in the 24-well plates, and activated with 5 μg/ml anti-CD3 mAb (Clone#145-2C11, Cat#BE0001-1, Bio X Cell) and 2 μg/ml anti-CD28 mAb (Clone#37.51, Cat#BE0015-1, Bio X Cell), or 0.2 million/ml irradiated APCs and CBir1 peptide (ThermoFisher Scientific) in the presence or absence of acetate (10 mM, Sigma Aldrich), propionate (0.5 mM, Sigma Aldrich), or butyrate (0.5 mM, Sigma Aldrich), under neutral (without exogenous cytokines), Th1 (10 ng/ml IL-12), Th17 (15 ng/ml TGFβ, 30 ng/ml IL-6, 10 μg/ml anti-IFNγ mAb, 5 μg/ml anti-IL-4 mAb), or Treg (5 ng/ml TGFβ and 10 μg/ml anti-IFNγ mAb) polarization conditions. Cells were cultured at 37 °C with 5% CO$_2$. On day 5, the cell yield was about 2 million/ml.

**RNA sequencing.** Mouse splenic CD4$^+$ T cells were activated with anti-CD3 (Clone#145-2C11, Cat#BE0001-1, Bio X Cell) and anti-CD28 mAb (Clone#37.51, Cat#BE0015-1, Bio X Cell) in the presence or absence of butyrate (0.5 mM) under neutral or Th1 conditions for 48 h. Cellular RNA was extracted, qualified, and followed by library construction at Novogene using an NEBNext® Ultra RNA Library Prep Kit for Illumia®. Briefly, mRNA was enriched, purified, and randomly fragmented. After synthesis of the first-strand cDNA using random hexamers primers, the second-strand cDNA was generated by using a custom second-strand synthesis buffer (Illumina), dNTPs, RNase H, and DNA polymerase I. After completing the double-stranded cDNA library through several steps, including terminal repair, poly-adenylation, sequencing adaptor ligation, size selection, and PCR enrichment, the 250–350 bp insert libraries were quantified by quantitative PCR. Qualified libraries were sequenced on an Illumina Novaseq Platform using a paired-end 150 run. Data were analyzed using Novosmart.

**Quantitative real-time PCR.** Total RNA was extracted from CD4$^+$ T cells by using Trizol, and reverse-transcribed to cDNA by using qScript cDNA Synthesis Kit (P/N#84035, Quantabio). Quantitative real-time PCR was performed for analysis of gene expression by using SYBR Green Gene Expression Assays (Cat#1725124, Bio-Rad). All the primers were ordered from Integrated DNA Technologies, and listed in Supplementary Table 2.

**ELISA.** IL-22, IL-6, and TNFα production were measured using ELISA MAX$^{TM}$ Deluxe Sets from Biolegend (IL-22, Cat#436304; IL-6, Cat#431304; TNF-α, Cat#430904). Microplate wells were incubated with capture antibodies overnight at 4 °C, and blocked with 1% BSA for 1 h at room temperature. Samples were incubated in the wells for 2 h, and followed by the addition of detection antibodies. Horseradish peroxidase-labeled streptavidin was then incubated in the wells for 30 min. After adding TMB substrate, cytokines concentrations were measured at 450 nm using BioTek Gene5 instrument.

**Flow cytometry.** Cells were stimulated with ionomycin (750 ng/ml, Invitrogen) and phorbol-12-myristate 13-acetate (50 ng/ml, Sigma Aldrich) for 2 h, then

7 μl/ml brefeldin (BD Biosciences) was added for another 3 h. Cells were first incubated with anti-CD16/32 for Fc block (1:10, Clone#93, Cat#101302, Biolegend), and then stained with Live/dye using LIVE/DEAD™ Fixable Near-IR Dead Cell Stain Kit (Cat#L10119, Invitrogen), followed by surface staining with mouse CD4 (1:200, Clone#RM4-5, Cat#100544, Biolegend) or human CD4 antibodies (1:200, Clone#A161A1, Cat#357409, Biolegend). Then, cells were fixed and permeabilized using the Foxp3/Transcription Factor Fixation/Permeabilization set (Cat#00-5523-00, ThermoFisher). Intracellular cytokines were then stained with different antibodies purchased from Biolegend or Invitrogen. Cytokine and transcription factor levels (anti-IFN-γ, 1:100, XMG1.2, #505806, Biolegend; anti-IL-17, 1:200, Clone#TC11-18H10.1, Cat#506922, Biolegend; IL-10, 1:200, Cat#JES5-16E3, Cat#505010, Biolegend; anti-Foxp3, 1:200, Clone#1H8PWSR, Cat#12-7221-82, Invitrogen; anti-IL-22, 1:200, Cat#FJK-16S, Cat#17-5773-82, Invitrogen) were shown as percentage of parent cells or absolute positive number cells in total spleen/MLN/lamina propria (LP) cells.

For staining with phosphorylated Stat3, mTOR, and S6 ribosomal protein, the cells were stained with Live/dye and anti-CD4 antibody (1:200, Clone#RM4-5, Cat#100510, Biolegend), and fixed by using IC Fixation Buffer (Cat#00-8222-49, ThermoFisher) for 30 min, followed by resuspension in 100% methanol for 1 h. After washing, cells were stained with anti-pStat3 (1:200, Clone#13A3-1, Cat#651004, Biolegend), anti-pmTOR (1:200, Clone#MRRBY, Cat#46-9718-42, Invitrogen), anti-pS6K1 antibody (1:200, Cat#8520, Cell Signaling Technology).

All the events were collected by BD FACS Diva software and analyzed using Flowjo. All the gating strategies were included in Supplementary Fig. 15.

**ILC culture and staining.** CD4$^+$ T cells-depleted splenic cells were cultured with IL-23 (20 ng/ml, Biolegend) with or without butyrate (0.5 mM) for 16 h, and then stimulated with ionomycin (750 ng/ml) and phorbol-12-myristate 13-acetate (50 ng/ml) for 2 h, followed by addition of brefeldin (BD Biosciences) for another 3 h. After Fc blocking, cells were stained with surface markers (PE/Cy7-anti-Thy1, 1:200, Clone#30-H12, Cat#105326; FITC-lineage (CD3, Clone#145-2C11, Cat#100306; CD11b, Clone#M1/70, Cat#101206; CD11c, Clone#N418, Cat#117306; B220, Clone#RA3-6B2, Cat#103206; F4/80, Clone#BM8, Cat#123108; NK1.1, Clone#PK136, Cat#108706; and Gr1, Clone#RB6-8C5, Cat#108419), 1:100), which were purchased from Biolegend. Cells were permeabilized using Foxp3/ Transcription Factor Fixation/Permeabilization set (Cat#00-5523-00, Thermo-Fisher), followed by intracellular staining (anti-IL-22, 1:200, Clone#1H8PWSR, Cat#12-7221-82, Invitrogen; anti-IL-17, 1:200, Clone#TC11-18H10.1, Cat#506916, Biolegend). For ILC3, cells were also stained with RORγt (1:200, Clone#B2D, Cat#17-6981-82, Invitrogen).

All the events were collected by BD FACS Diva software. IL-22 and IL-17 production in ILCs (Thy1$^+$ Lineage$^-$ cells), or in ILC3 (Thy1$^+$ Lineage$^-$ RORγt$^+$ cells) was analyzed using FlowJo. Gating strategies are included in Supplementary Fig. 15.

**Dendritic cells and NKT cell staining.** For dendritic cell (DC) staining, intestinal lamia propria cells were stimulated with ionomycin (750 ng/ml, Invitrogen) and phorbol-12-myristate 13-acetate (PMA, 50 ng/ml, Sigma Aldrich) for 2 h, then 7 μl/ml brefeldin (BD Biosciences) was added for another 3 h. Cells were first incubated with anti-CD16/32 for Fc blocking (1:10, Clone#93, Cat#101302, Biolegend), and then stained with Live/dye using LIVE/DEAD™ Fixable Near-IR Dead Cell Stain Kit (Cat#L10119, Invitrogen), followed by surface staining with anti-

CD11c antibody (1:100, Clone#N418, Cat#117306, Biolegend). Then, cells were fixed and permeabilized using the Foxp3/Transcription Factor Fixation/Permeabilization set (Cat#00-5523-00, ThermoFisher), followed by staining anti-IL23p19 antibody (1:200, Clone#fc23cpg, Cat#50-7023-82, Invitrogen).

For NKT cell staining, intestinal lamia propria cells were treated with 20 ng/ml IL-23 for 16 h, and then stimulated with 750 ng/ml ionomycin and 50 ng/ml PMA for 2 h, then 7 μl/ml brefeldin (BD Biosciences) was added for another 3 h. Cells were first incubated with anti-CD16/32 for Fc block (1:10, Clone#93, Cat#101302, Biolegend), and then stained with Live/dye, followed by surface staining with anti-NK1.1 (1:100, Clone#PK136, Cat#108706, Biolegend) and anti-TCRβ antibodies (1:200, Clone#H57-597, Cat#109230, Biolegend). The cells were then fixed and permeabilized, followed by staining anti-IL22 antibody (1:200, Clone#1H8PWSR, Cat#12-7221-82, Invitrogen).

All the events were collected by BD FACS Diva software and analyzed using Flowjo. All the gating strategies were included in Supplementary Fig. 15.

**HDAC activity assay**. CD4$^+$ T cells were with anti-CD3 (Clone#145-2C11, Cat#BE0001-1, Bio X Cell) and anti-CD28 mAb (Clone#37.51, Cat#BE0015-1, Bio X Cell) under Th1 conditions in the presence or absence of butyrate (0.5 mM) or TSA (10 mM) for 24 h. Cells were collected, and nucleoprotein was extracted by using NE-PER Nuclear and Cytoplasmic Extraction Reagents (Cat#78835, ThermoFisher). Then, nucleoprotein samples were incubated with HDAC Green substrate (Cat#13601, AAT Bioquest) for 40 min at 37 °C, followed by measuring the fluorescence intensity at excitation/emission (490/525 nm).

**Western blot**. CD4$^+$ T cells were activated with anti-CD3 (Clone#145-2C11, Cat#BE0001-1, Bio X Cell) and anti-CD28 mAb (Clone#37.51, Cat#BE0015-1, Bio X Cell) in the presence or absence of butyrate (0.5 mM). Cells were collected at 6 h or 24 h. Proteins were extracted by using a radio-immunoprecipitation buffer containing protease inhibitor cocktail, phosphatase inhibitor cocktail, and PMSF. Protein concentrations were determined by using a Pierce BCA Protein Assay Kit (Cat#23225, ThermoFisher Scientific). 10 μg protein of each sample was loaded into the NuPAGE 4–12% Bis-Tris mini gels (Life Technologies), and separated electrophoretically. Then, proteins were transferred to the PVDF membrane. After blocking with 5% non-fat milk, the membrane was incubated with primary antibodies overnight at 4 °C. Next, secondary antibodies (1:2000) were incubated with the membrane for 1 h at room temperature. After incubating with substrate, blots were detected by using the ImageQuant™ LAS 4000 biomolecular imager. Primary antibodies:anti-phosphorylated Stat3, 1:2000, Y705, Cat#9145, Cell Signalling Technology; anti-Stat3, 1:2000, D3Z2G, Cat#12640, Cell Signaling Technology; anti-phosphorylated mTOR, 1:2000, S2448, Cat#2971, Cell Signaling Technology; anti-mTOR, 1:2000, Cat#2972, Cell Signaling Technology; anti-HIF1α, 1:1000, D2U3T, Cat#14179, Cell Signaling Techonolgy; anti-HIF2α, 1:1000, Cat#AF2997, R&D Systems; anti-AhR, 1:1000, Cat#AF6697, R&D Systems; anti-βactin, 1:2000, D6A8, Cat#8457, Cell Signaling Technology. Uncropped gels were provided in Source Data file.

**HIF1α activity assay**. CD4$^+$ T cells were activated with anti-CD3 (Clone#145-2C11, Cat#BE0001-1, Bio X Cell) and anti-CD28 mAb (Clone#37.51, Cat#BE0015-1, Bio X Cell) in the presence or absence of butyrate (0.5 mM) for 24 h. Cells were collected, and nucleoprotein was extracted by using NE-PER Nuclear and Cytoplasmic Extraction Reagents (Cat#78835, ThermoFisher), and HIF1α activity in samples was measured using HIF1α Transcription Factor Assay Kit (Cat#ab133104, Abcam). Nuclear protein was diluted with Complete Transcription Factor Binding Assay Buffer, and then incubated in the wells of Transcription Factor HIF1α plate overnight at 4 °C. Primary HIF1α antibody was then incubated in the wells for 1 h at room temperature, followed by the addition of the second goat anti-rabbit HRP antibody for another 1 h. Then, Transcription Factor Developing Solution was then incubated in the wells for 30 min at room temperature. After adding the Stop solution, HIF1α activity levels were measured at 450 nm using BioTek Gene5 instrument.

**Lentivirus transduction and luciferase assay**. RAW 264.7 cells were seeded in 6-well plates (1 × 10$^6$ per well). On day 2, cells were transduced with Xenobiotic Response Elelment (XRE/AhR) Luciferase Reporter Gene Lentivirus particles (SKU#LTLR031, g&p biosciences) at 10 MOI in the media containing 10 μg/ml hexadimethrine bromide. After 48 h, cells were treated with or without butyrate (0.5 mM) for 24 h. Cells were collected and lysed using lysis buffer (Promega), and then transferred to a 96-well plate. Relative fluorescent units (RFU) were then measured after injecting Luciferase Assay Reagent (E1500, Promega).

**Hypoxia cell culture**. Mouse splenic CD4$^+$ T cells were activated with anti-CD3 (Clone#145-2C11, Cat#BE0001-1, Bio X Cell) and anti-CD28 mAb (Clone#37.51, Cat#BE0015-1, Bio X Cell) in the presence or absence of butyrate (0.5 mM) under normal oxygen circumstance (20% oxygen) or hypoxia condition (3% oxygen).

**Chromatin immunoprecipitation assay**. Mouse splenic CD4$^+$ T cells were activated with anti-CD3 (Clone#145-2C11, Cat#BE0001-1, Bio X Cell) and anti-CD28 mAb (Clone#37.51, Cat#BE0015-1, Bio X Cell) in the presence or absence of butyrate (0.5 mM) for 2 days. CHIP was performed using a Chip-ITtm Express Magnetic Chromatin Immunoprecipitation Kits (Cat#53008, Active Motif). Briefly, cells were fixed in 1 ml 1% formaldehyde on a shaker for 7 min at room temperature. After adding Glycine Stop-Fix Solution, the samples were shaken for 5 min at room temperature. After cell lysis, nuclei pellets were resuspended in the shearing buffer for chromatin shearing by sonication. After assessing the efficiency of shearing, 25 μg sheared chromatin was used for the CHIP reaction by incubating with anti-IgG (DA1E, Cat#3900Cell Signaling Technology, 3 μg/sample), anti-HIF1α (Cat#NB100-134, Novus Biologicals, 3 μg/sample), anti-H3k9ac (Cat#391372, Active Motif, 10 μl/sample), or anti-H3K9me3 (Cat#39161, Active Motif, 5 μl/sample), together with protein G magnetic beads overnight at 4 °C. After a series of washes magnetic beads, chromatin elution, reversing cross-links, and proteinase K treatment, DNA was purified by using phenol/chloroform TE. Real-time PCR was used for analysis of the amount of immunoprecipitated DNA. The primers used were listed in Supplementary Table 2. Fold Enrichment Method was used to normalize CHIP-qPCR data. The average Ct values for specific IgG IP were determined, and the average Ct values for anti-IgG Ab IP were determined as background signal. Then, the average CT values for specific IP were subtracted from the average Ct for anti-IgG Ab. Finally, the fold enrichment was calculated as $2^{-\Delta\Delta Ct}$.

**Butyrate administration in mice in vivo**. For investigating the effect of butyrate in inducing IL-22 in vivo, mice were administrated with 200 mM butyrate in drinking water for 3 weeks. For *C. rodentium* infection, mice were administrated with 200 mM butyrate in drinking water for 10 days post oral infection. For DSS colitis model, mice were administrated with 200 mM butyrate in drinking water containing 1.65% DSS for 7 days, and then 200 mM butyrate in drinking water for another 3 days.

***Citrobacter rodentium* infection**. WT and *Il22*$^{-/-}$ mice were orally infected with *C. rodentium* (strain DBS100, ATCC, 5 × 10$^8$ colony-forming unit (CFU)/mice) on day 0. Mouse weights were monitored daily. Feces were collected on day 7 for analysis of *C. rodentium* clearance. On day 10, mice were killed, and the liver was homogenized for analysis of *C. rodentium* dissemination, and the colon and cecum were used for analysis of colitis severity.

For depleting CD4$^+$ T cells and ILCs in vivo, mice were administered control anti-IgG antibody (25 mg/kg, Clone#C1.18.4, Cat#BP0085, Bio X cell), anti-CD4 antibody (25 mg/kg, Clone#GK1.5, Cat#BP0003-1, Bio X cell), or anti-Thy1 (25 mg/kg, Clone#30-H12, Cat#BE0066, Bio X cell) i.p. every the other day.

**Fecal and liver *Citrobacter rodentium* measurement**. Feces were collected 7 days post *C. rodentium* infection. Feces were weighed, resuspended, and homogenized. Fecal supernatants were plated at a series of concentrations into MacConkey agar-plates (BD Biosciences) at 37 °C overnight, and colony-forming units (CFU) were counted. Liver was obtained when mice were killed on day 10 post-infection, weighed, homogenized, and plated at a series of concentrations onto MacConkey agar-plates at 37 °C overnight, and CFU were counted.

**Dextran sulfate sodium colitis**. WT and *Il22*$^{-/-}$ mice were administrated with 1.65% DSS (Gojira FC) in drinking water for 7 days, and switched to untreated water for another 3 days. Mouse weights were monitored daily. Mice were killed on day 10, and the colon and cecum were collected for assessment of histopathology.

**Human peripheral blood CD4$^+$ T cell isolation and culture**. CD4$^+$ T cells were isolated from peripheral blood of healthy volunteers, CD patients, and UC patients using anti-human CD4 magnetic particles (Cat#557767, BD Biosciences). Cells were activated with 5 μg/ml anti-CD3 (Clone#HIT3a, Cat#300302, Biolegend) and 2 μg/ml anti-CD28 (Clone#CD28.2, Cat#302901, Biolegend) in the presence or absence of butyrate (0.5 mM).

**Quantification of fecal butyrate by LC–MS**. Mice fecal samples were extracted with 80% Isopropanol using cell disruptor (Sonibeast-BioSpec). Zirconia/Silica Beads (1 mm) were added to each tube prior to extraction. Each sample was pulsed for 20 s followed by 1 min on ice. This was done trice for a total of 60 s pulse. The samples were extracted in a total of 3 ml of 80% Isopropanol. LC–MS analysis was performed after derivatization of the samples with 3-NPH. Standard solutions of analyte (Butyric) were run at 10 different concentrations. D7-butyric acid was used as an internal standard and added to each sample and standard prior to derivatization. Data were analyzed using Multiquant (SCIEX). The concentrations were normalized to mg of fecal sample.

**Histopathology assessment**. Colon and cecum samples were collected, rolled into a Swiss Roll, and fixed in 10% buffered formalin in the tissue cassettes. Samples were then paraffin-embedded, sliced, and stained with Haemotoxylin and Eosin (H&E). Pathology scores were assessed by the following criteria. For DSS-induced colitis model, the colitis severity was quantified by cell infiltrate (normal, 0; mild in mucosa, 1; moderate in mucosa, 2; marked in mucosa, 3; moderate/severe in mucosa and submucosa, 4; transmural, 5) and architecture (no erosion, 0; focal

erosion, 1; focal ulceration, 2; extended ulcerations, 3). For *C. rodentium* infection model, the disease severity was quantified by epithelium change (normal, 0; mild, 1; moderate, 2; severe, 3), lamina propria inflammation (normal, 0; mild, 1; moderate, 2; severe, 3), affected area (none, 0; 0–25%, 1; 25–50%, 2; >50%, 3), and server marker (none, 0; mild submucosal inflammation or <5 crypt abscesses, 1; mild submucosal inflammation and <5 crypt abscesses, 2; severe submucosal inflammation or >5 crypt abscesses or crypt branching, 2; severe submucosal inflammation and >5 crypt abscesses or crypt branching, 3; ulceration or extensive fibrosis, 3).

**Statistical analysis.** Student's unpaired or paired *t*-test was used to compare two groups when data were normally distributed. The nonparametric Mann–Whitney *U* test was used to measure the difference between two groups in which data were not normally distributed. For comparing more than two groups, one-way ANOVA was performed. All the tests were two-sided. All the analysis was performed by using Graphpad Prism 8.0 software. All the data were presented as mean ± SD. $*p < 0.05$, $**p < 0.01$, $***p < 0.001$.

**Reporting summary.** Further information on research design is available in the Nature Research Reporting Summary linked to this article.

## Data availability

RNA-seq data have been deposited in GEO database under the accession number GSE139631. All other data supporting the findings of this study are available within the article and its supplementary information files or from the corresponding author upon reasonable request. The source data underlying Figs. 1a–i, 2a–h, 3a–d, 4a–m, 5a–j, 6b–e, 7a–h, 8a–f, 9a–f, and Supplementary Figs. 1c–e, 1h–j, 2, 3a, d, 4, 5a, b, 8a–j, 9a–h, 10a–f, 11a–f, 12a–l, 13a–c, 14a–f. Source data were provided as a Source Data file. Source data are provided with this paper.

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

## Acknowledgements

This work was supported by NIH Grants DK105585, DK112436, and DK125011, and University of Texas System STARs award. We appreciate Dr. Sherry Haller of The University of Texas Medical Branch for proof reading the manuscript. Images in Figs. 6a and 10 were created with BioRender.com.

## Author contributions

Conceptualization: W.Y. and Y.C.; methodology: W.Y., T.Y., S.M.D., and Y.C.; investigation: W.Y., T.Y., X.H., A.B.J., L.X., Y.L., S.Y., S.M.D., and Y.C.; resources: J.S., F.P., J.Z., W.Z., Z.L., C.L.M., and N.S.; writing—original draft preparation: W.Y. and Y.C.; writing—review and editing: W.Y., J.S., F.P., and Y.C. with input from all other authors; supervision: Y.C.; funding acquisition: Y.C.

## Competing interests

The authors declare no competing interests.
