## [Peer Review File · Nature Communications]

Reviewers' comments:

Reviewer #1 (Remarks to the Author):

In this manuscript, Yang et al. report that SCFA promotes IL-22 production in T cells and ILCs. Furthermore, they demonstrate that SCFA promoted HIF1 α binding to the IL22 promoter by inhibiting HDAC in T cells. They also demonstrate that SCFA enhanced oxygen consumption and glycolysis in T cells, which augments IL-22 production in T cells. Supplementation of SCFA protects mice from C Rodentium-induced and DSS-induced colitis by inducing IL-22. Finally, they demonstrate that butyrate induces IL-22 production in human T cells.

Overall, the authors provided an impressive body of evidence demonstrating that SCFA promotes IL-22 production in T cells and ILCs. Butyrate upregulation of IL-22 in T cells and ILCs is mediated by the transcription factor HIF1 α and AHR. Several issues described below need to be addressed to strengthen the manuscript. In addition, the data falls short in investigating the relation between HIF1 α and glycolysis in the setting of T cells treated by SCFA. The relationship between the SSFA effects on HDAC1 inhibition/HIF1 α activation, AHR activation, and the metabolic-dependent effects on IL22 expression is unclear.

Major Comments:

1. Which cell type(s) is required for the butyrate-induced effects of IL22: ILCs, T cells or both in vivo? In vivo depletion/transfer experiments are needed.
2. IL-22+ T cells and ILCs increased in mice following butyrate administration. What is the relative expression of IL22 in the different immune compartments in vivo (Th1, Th17, Treg, ILC3, or all)? This can be accomplished by more detailed staining.
3. For the in vitro T cells experiments, what's the efficiency of each treatment? For example, under Th1 conditions, what's the percentage of Th1 cells at the end of treatment? Does butyrate change the percentage?
4. Given GPR43 appears not to play a role, how do the authors believe SSFA is working in ILCs and T cells? This should be addressed at least in the discussion
5. The mechanistic studies were largely performed with T cells. The authors claim SCFA promotes IL-22 production in ILCs through identical mechanisms. More data is needed.
6. The authors claim a role for both HIF and AhR in the production of IL-22 in response to SCFA. However, they only comment on HIF binding sites in the promoter of IL-22. Are their AhR binding sites?
7. Do Stat3 and mTOR inhibitors inhibit each other's signaling. What is the status of IL22-dependent mTOR phosphorylation in the absence of Stat3?
8. Total STAT3 and mTOR WB are needed for Fig. 4. Quantification of the phosphorylated band and total protein band and calculation of the ratio is needed.
9. In Figure 5 if Butyrate promotes HIF1 α binding to IL-22 promoter depends on the inhibition of HDAC, stronger inhibition of HDAC should induce higher IL-22 production. TSA showed stronger inhibition of HDAC than butyrate (Fig.5F), but induced lower IL-22 (Fig.5G-H).
10. The unique and non-redundant relationship between the SSFA effects on HDAC1 inhibition/HIF1 α activation, AHR activation, and the metabolic-dependent effects on IL22 expression is unclear. Reprogramming of the glucose metabolic pathway is a known effect of HIF1 α activation. The metabolic effects while interesting detracts some from the manuscript.
11. CD patient and healthy control T cells show similar or even higher IL-22 production both in homeostasis and after butyrate treatment. The use of butyrate in CD may therefore not be beneficial. Given the murine models assessed are more reminiscent of UC, have experiments been performed with UC patient cells?
12. An overall schematic summarizing the mechanism would be helpful.

Minor Comments:

1. The quantification panels of Fig. 2D and Fig. 2E are identical. Please correct.

2. Fig. 7 states the dose of *C. Rodentium* as 1×10^9 , while in the results section, the dose is given as 5×10^8 ? Please clarify.
3. Fig. 8G was mentioned in the results section but G is not present in the figure. This needs to be updated.
4. Please indicate the concentration of Etomoxir used.
5. ILCs are innate lymphoid cells and not innate lymphocyte cells.
6. Please use the correct formatting for knockout mice e.g. italics for mouse genes and superscript for flox.
7. Some subtitles are needed for the figure panels to orient the reader. For example, Fig.1A and Fig.1E are so similar, readers cannot easily discern the difference between T cells and Th1 Cells as labeled.
8. Figure legends for Fig. S1 (F-G) are the same as Fig. S1 (A-B).
9. "One representative of three independent experiments was shown" is wrote in every figure legend. Is the true for each Figure?
10. Gating strategy for the FACS data is necessary.

Reviewer #2 (Remarks to the Author):

In the manuscript entitled "Gut microbiota metabolite short-chain fatty acids protect the intestine from inflammation through promoting CD4+ T cell and innate lymphocyte cell IL-22 production" the authors provide novel insight into the regulation of microbiota-induced IL-22 production by T cells and ILCs in the intestine during colitis. Several major observations are presented: 1) SCFAs promote IL-22 production via inhibition of HDAC activity; 2) SCFAs induce IL-22 by promoting AhR and HIF1a expression; 3) HIF1a directly binding to the IL-22 promoter and SCFAs increase HIF1a binding to the IL-22 promoter; 4) SCFAs enhance oxygen consumption and glycolysis; 5) SCFA supplementation protects from Citro- and DSS-induced colitis in association with enhanced IL-22; and 6) SCFAs promote T cell IL-22 production from patients with Crohn's disease. Understanding how IL-22 is regulated during intestinal inflammation is an important and potentially clinically relevant area of investigation.

Overall, the manuscript is well-written and data presented are of high quality and very thorough. Addressing the following points may further strengthen this important study:

Individual experiments with only 3 replicates appear low for many of the studies.

To accompany Fig 1 a list of the top 100 or so induced and repressed genes and their levels of induction/repression would be helpful to the reader.

In Fig 2B/C it should be "(pg/ml)" instead of "(pg)".

In Fig 3D/4A/B please label (or reference in the legend/text) the MW of the proteins.

How are SCFAs directly sensed by cells to inhibit HDAC activity if not via GPR43?

Data in Fig 6 showing that SCFAs enhance oxygen consumption and glycolysis does not appear well integrated into the rest of the manuscript. Additional rationale, discussion appears warranted.

In Fig 6 D is should be "ECAR" not "EACR".

In the abstract and main text it is stated that SCFAs promote T cell IL-22 production from patients with Crohn's disease. This is technically correct, but SCFAs also promote T cell IL-22 production from HC. Conclusions should be modified accordingly.

Throughout the manuscript the authors refer to "T cells" when they are assessing CD4+ T cells. It is recommended that they are referred to as "CD4+ T cells" to be accurate.

Reviewer #3 (Remarks to the Author):

The manuscript by Yang et al, focuses on the potential that gut microbiota derived short chain fatty acids (SCFA) promote IL-22 production by CD4 T cells and ILC3s, through an array of different pathways and mechanisms including inhibiting HDAC activity, as well as increasing HIF-1 α and AHR expression. Moreover, SCFA supplementation protects the colon from infection and colitis in an IL-22 dependent manner.

While the study concept is interesting, the manuscript has several weaknesses that reduce my enthusiasm. In particular the ability of SCFA to modulate immune cell function in vitro has been well described previously (inducing T regs, causing apoptosis of T cells etc), so while the focus on IL-22 is interesting, the novelty, and complexity of the mechanisms is worrying. While the authors demonstrate that adding high levels of butyrate to immune cells alters function in vitro, the in vivo relevance is unclear, since although the authors show modest effects by gavaging mice with butyrate – that is not the normal source of butyrate and other SCFA (ie. they are made in the colon). There is little evidence that SCFA made in the colon actually affects systemic immunity. As such, they should perhaps repeat their studies in germfree mice, and assess how the microbial based (from the colon) SCFA production alters immune cells. Moreover, the use of C_{Bir} T cells is problematic (as they are highly artificial).

Major issues.

(1) The premise of the paper is that SCFAs promote IL-22 production from CD4 T cells and ILCs to protect against intestinal inflammation. If that's the case why are splenocytes chosen over cells from the GALT or lamina propria? Wouldn't MLN also be a more appropriate source for cell isolation compared to the spleen? Do cells from these different sites respond differently to the SCFA? Do they polarize differently? Do they proliferate at different rate?

(2) While I understand the reason for testing the C_{Bir}1 transgenics, they do pose problems. Having a transgenic TCR completely changes how thymocytes develop and what central and peripheral tolerance mechanisms are in play. The transgenic TCR in question is specific to commensal flagellin, which is abundant in the GI tract. Are these mice germ-free? If not, the T cells would likely have already encountered their cognate antigen and undergone changes such as deletion, anergy, or worse epigenetic reprogramming (Greenberg, 2012). I think it would be better for the authors to focus on WT cells, but examine them from different sites, including the gut.

(3) In this report, an epigenetic mechanism driven by Hif1 α was proposed. If this is the case, then perhaps the effect of oxygen tension on the capacity of T cells to produce IL-22 should be taken into account, and only T cells from the GALT should be used instead of the spleen? Cladwell et al (Cladwell, 2011) seem to suggest that spleen is rather hypoxic compare to the GALT.

(4) As for the Th polarizing conditions, does the addition of anti IFN γ /IL-4 cytokine antibodies affect the outcome of polarization and IL-22 production? Does the addition of IL-23 affect Th17/ILC polarization/response and IL-22 secretion? IL-2 is critical for several lineages of T cells, it is very surprising to see that it's not mentioned even once in this article.

(5) Please provide more detail on how the T cells were expanded (ie, how many were activated, what CD3/CD28 clones were used in what kind of plates/culturing container, at what cellular concentration/number before during and after the expansion). Also, please provide some CD44 stain data on the T cells before and after the expansion. These data would confirm that the T cells are viable and hence the differential cytokine expression seen is purely due to the presence of

butyrate.

Minor issues

Figure 2. Dendritic cells are an important source of IL-23 and play critical roles in the polarization of T helper cells. Are there any changes in IL-23 secretion by the dendritic cells in the MLN and lamina propria after 3 weeks of butyrate in drinking water?

Also, please provide data on the intestinal microbiota. Butyrate can modulate oxygen availability, and since the intestinal microbiota can shift drastically based on oxygen availability. It would be helpful to determine whether the change in IL-22 production is due to butyrate, the change of microbiota or a combination of both.

Downregulation of CD4 or CD8 can be a sign of T cell activation. How are these T cells gated in figure 2D? What would the data look like if the plot were not gated on the CD4 high cells first? Are there any CD4⁻ cells secreting IL-22 in these sites? What are the absolute numbers for these IL-22 secreting cells?

IL-22 can be secreted by NKT cells, which can be protective in intestinal inflammation. Does butyrate enhance IL-22 secretion from NKT cells as well?

Figure 3. Is butyrate activating Hif1a? Can butyrate activate AHR? There seem to be evidence that butyrate can activate AHR in human intestinal epithelial cells (Marinelli, 2019), is this true in mice as well?

What is the rationale of YC-1 as Hif1a inhibitor? There were reports that YC-1 also acts on Hif2a. Will other Hif1a inhibitors produce the same effect? Although Hif1a expression was shown, what about its activity? How much Hif1a activity is YC-1 blunting? How about Hif2a? For figure 3D, why not show both Hif1a Hif2a and AhR western blot? DMOG treated T cells would be a nice control to have (please see reference 34).

Please provide the rationale why the CBir1 transgenic were used in one part of the figure and B6 for the other part? According to the methods section, they were all stimulated by PMA/Ionomycin, which defeats the purpose of a TCR transgenic. Were the CBir1 cells responsive to the YSNANILSQ peptide? If so, how did they respond to different dosages of the peptide? Would be nice to see flow data on the IFN gamma and IL-22 readout.

For figures 3E and F, please show flow cytometry data. Does treatment of inhibitors affect cell viability?

For figure 3G, it appears that the majority of IL-22 producing cells in the butyrate treated WT cells are low IL-22 producers. Would the inclusion of IL-22 FMO/ISO help determine whether these are indeed IL-22 secreting cells? Also, it is rather difficult to see these dots. It would be preferable to show the plots in low-resolution mode so that it's easier to see the dots but more preferable that more events were acquired (50-100,000 CD4⁺ T cells should suffice). Also please include Hif2a^{-/-} T cells in the data if Hif2a activity is indeed affected by the inhibitor.

The figure legend of (H-I) "...Th1 conditions with or without (0.5mM) under normxic" does not make sense, is the word missing butyrate and the typo normoxic?. Further, what is the justification of using 3% oxygen (please see Cladwell et al, 2011)? Would a T cell in the lamina propria/MLN experience such concentration of oxygen (please see Carreau 2011, Espey, 2013)?

Figure 4. For 4A&B, it would be nice to have density quantification and stats. The effect of butyrate on mTOR shown in 4B seemed minimal, would phosphor flow be a better option? For the phosphor

flow data, please indicate what fluorochrome and clone in the methods section.

For 4C, how were the cells stained and gated? Please provide more information in the methods section. Please also provide MFI and stats.

For 4H why was there a drop in IL-22 in the WT? This look very different compared to 3G 2D 2E and 1H. Also, the figure legend for 4H is confusing. Can treatment with butyrate alone result in IL-22 production?

Figure 5. If IL-22 secretion was assessed on day 5 post activation, what is the justification of performing ChIP on IL-22 promoter on day 2?

5A: To make the claim that these HRE binding sites are indeed binding Hif1a one need to first list the DNA sequence, then map for local topology, and then follow with experiments showing that Hif1a is indeed binding on these sites.

For the ChIP experiments, please elaborate on how the fold enrichment was calculated in your methods section and provide cytokine data for day 2.

Figure 6. Please show metabolism data on GPR43^{-/-} and STAT3^{-/-} CD4 T cells. The difference seen could be due to additional activation signal induced by butyrate through GPR43 and STAT3.

Figure 7. For 7B, the histology shown is too small to really determine what is going on. Please include pathogen burdens. marginal, please include WT uninfected. For flow cytometry data, please provide data in numbers not percentages. The numbers of events on the plots are not even, making it difficult to analyze.

Figure 8. For 8B. It is CD4 T cells that were isolated? If so, how were they stained and gated? Why choose SSC over CD4? Please provide clone names and fluorochrome for antibodies and gating strategy in figure legend or in methods. Also for the CD patients, how were they treated for their disease?

Reply to comments of Reviewer #1

1. Which cell type(s) is required for the butyrate-induced effects of IL22: ILCs, T cells or both in vivo? In vivo depletion/transfer experiments are needed.

Response: We appreciate the reviewer's excellent comments and suggestion. To investigate whether ILCs, T cells, or both cells are required for the effect of butyrate on intestinal inflammation, we used anti-CD4 mAb to deplete CD4⁺ T cells, and anti-Thy1 mAb to deplete CD4⁺ T cells and ILCs in mice, as there are currently no antibodies to specifically deplete ILCs only. We found that both ILCs and CD4⁺ T cells are important in protecting the intestines from inflammation induced by butyrate *in vivo*, which are included in the revised manuscript now (**revised Fig. 7G-H**).

2. IL-22⁺ T cells and ILCs increased in mice following butyrate administration. What is the relative expression of IL22 in the different immune compartments in vivo (Th1, Th17, Treg, ILC3, or all)? This can be accomplished by more detailed staining.

Response: We appreciate the reviewer's excellent comments, and have performed the experiment as suggested. We examined the IL-22 expression in lamina propria IFN γ ⁺, IL-17⁺, or Foxp3⁺ CD4⁺ cells, and in lamina propria Ror γ ⁺ ILCs from mice before and after 3-week butyrate treatment. The new data is included in the revised manuscript now (**revised Fig. 2F and 2H**).

3. For the in vitro T cells experiments, what's the efficiency of each treatment? For example, under Th1 conditions, what's the percentage of Th1 cells at the end of treatment? Does butyrate change the percentage?

Response: We appreciate the reviewer's excellent comments. According to the methods described in manuscript, the percentages of IFN⁺ Th1 cells were around 80-90%, the percentages of IL-17⁺ Th17 cells were around 50-60%, and the percentage of Foxp3⁺ Treg cells were around 40-50%, under Th1, Th17, and Treg conditions, respectively. Additionally, butyrate promoted Treg differentiation, and inhibited Th17 cell differentiation, which was consistent with previous reports. However, butyrate did not affect Th1 differentiation, which might due to high percentages of IFN- γ ⁺ T cells under Th1 conditions. Please see **new fig. S2**.

4. Given GPR43 appears not to play a role, how do the authors believe SCFA is working in ILCs and T cells? This should be addressed at least in the discussion.

Response: We appreciate the reviewer's excellent comment. In addition to activation of G-protein-coupled receptors (GPCRs), SCFA can enter into cells through passive diffusion or carrier-mediated transportation to function as HDAC inhibitors to regulate cell differentiation and function [1, 2]. In our study, we found butyrate promoted IL-22 production through their inhibition on HDAC activity but not GPR43. Following the reviewer's suggestion, we now investigated whether GPR41 and GPR109a, the other two receptors for SCFAs, mediate the butyrate induction IL-22 in CD4⁺ T cells and ILCs. We found that similar to butyrate, GPR41 specific agonist promoted IL-22 production in both CD4⁺ T cells and ILCs. However, deficiency in CD109a did not affect butyrate induction IL-22, indicating that in addition to inhibition of HDAC, butyrate induction of IL-22 is also GPR41 dependent, at least partially. These data are included in the revised manuscript now (**new Fig. 3, fig. S8, S9A-C**).

- [1] Sun M, Wu W, Liu Z *et al.* Microbiota metabolite short chain fatty acids, GPCR, and inflammatory bowel diseases. *Journal of gastroenterology* **52**, 1-8 (2017).
- [2] Correa-Oliveira R, Fachi JL, Vieira A *et al.* Regulation of immune cell function by short-chain fatty acids. *Clinical & translational immunology* **5**, e73 (2016).

5. The mechanistic studies were largely performed with T cells. The authors claim SCFA promotes IL-22 production in ILCs through identical mechanisms. More data is needed.

Response: We appreciate the reviewer's excellent comments. We now included the mechanistic data in ILCs in the revised manuscript (**fig. S9**). However, due to low cell yield of ILCs, we were not able to perform the CHIP assay as done in T cells.

6. The authors claim a role for both HIF and AhR in the production of IL-22 in response to SCFA. However, they only comment on HIF binding sites in the promotor of IL-22. Are their AhR binding sites?

Response: We apologize that we did not provide the information on AhR binding to *il22* promoter. Several AhR binding sites in *il22* promoter have been identified previously [1, 2, 3], we thus did not perform the experiments to verify the AhR binding sites.

- [1] Yeste A, Mascanfroni ID, Nadeau M *et al.* IL-21 induces IL-22 production in CD4⁺ T cells. *Nature communications* **5**, 3753 (2014).
- [2] Qiu J, Heller JJ, Guo X *et al.* The aryl hydrocarbon receptor regulates gut immunity through modulation of innate lymphoid cells. *Immunity* **36**, 92-104 (2012).
- [3] Bauche D, Joyce-Shaikh B, Fong J *et al.* IL-23 and IL-2 activation of STAT5 is required for optimal IL-22 production in ILC3s during colitis. *Science immunology* **5**, (2020).

7. Do Stat3 and mTOR inhibitors inhibit each other's signaling. What is the status of IL22-dependent mTOR phosphorylation in the absence of Stat3?

Response: We have done the experiments accordingly. We found that mTOR inhibitor moderately suppressed Stat3 activation, but Stat3 inhibitor did not affect mTOR phosphorylation. Addition of mTOR inhibitor further reduced butyrate-induced IL-22 production suppressed by Stat3 inhibitor in CD4⁺ T cells. These data are included in the revised manuscript now (**new fig. S13**).

8. Total STAT3 and mTOR WB are needed for Fig. 4. Quantification of the phosphorylated band and total protein band and calculation of the ratio is needed.

Response: We appreciate the reviewer's excellent comments. We have done the analysis accordingly, which is included in the revised manuscript now (**revised Fig. 5A and 5C**).

9. In Figure 5 if Butyrate promotes HIF1a binding to IL-22 promoter depends on the inhibition of HDAC, stronger inhibition of HDAC should induce higher IL-22 production. TSA showed stronger inhibition of HDAC than butyrate (Fig.5F), but induced lower IL-22 (Fig.5G-H).

Response: We appreciate the reviewer's excellent comments. According to the data from Figure 5, we modified the statement in manuscript as "Butyrate promoted CD4⁺ T cell production IL-22 at least partially through inhibiting HDAC". Additionally, as in response question #4, we also checked whether GPR41 and GPR109, the other two receptors for SCFAs, affect the butyrate induction IL-22 in CD4⁺ T cells and ILCs, and found GPR41 was at least partially involved, which are included in revised manuscript (**new Fig. 3A-D and fig. S8I**). Furthermore, HDAC

inhibitor and GPR41 agonist cooperatively promoted IL-22 production in T cells, in which IL-22 levels were similar as induced by butyrate (**new Fig. 3A-D**).

10. The unique and non-redundant relationship between the SCFA effects on HDAC1 inhibition/HIF1 α activation, AHR activation, and the metabolic-dependent effects on IL22 expression is unclear. Reprogramming of the glucose metabolic pathway is a known effect of HIF1 α activation. The metabolic effects while interesting detracts some from the manuscript.

Response: We appreciate the reviewer's careful reading through our manuscript. According to the comments of Reviewer #1 and also Reviewer #2, we removed the data regarding metabolic regulation in the revised manuscript.

11. CD patient and healthy control T cells show similar or even higher IL-22 production both in homeostasis and after butyrate treatment. The use of butyrate in CD may therefore not be beneficial. Given the murine models assessed are more reminiscent of UC, have experiments been performed with UC patient cells?

Response: We appreciate the reviewer's excellent comments. As suggested, we performed T cells from UC patients, which is included in the revised manuscript (**new Fig. 9**). It has been shown that several factors regulate IL-22 production in CD4⁺ T cells, including IL-6 and IL-1 β , which are increased in patients with IBD. Therefore, the relative high levels of IL-22 in CD patients may result from higher levels of proinflammatory cytokines. These relative high levels of IL-22 actually protect the host against bacteria attack and contribute to mucosal healing. It has been shown recently that anti-TNF therapy increased T cell IL-22 production to promote epithelial cell repair, which serves as one of the benefits for anti-TNF therapy [1]. Although we cannot definitely tell whether IL-22 is beneficial for IBD patients or not right now, clinical relevance of IL-22 to IBD has been well highlighted. Our data indicates that SCFAs possibly not only protect the normal humans but also protect the IBD patients from intestinal inflammation through upregulation of CD4 T cell IL-22 production.

[1] Fang L, *et al.* Anti-TNF Therapy Induces CD4⁺ T-Cell Production of IL-22 and Promotes Epithelial Repairs in Patients With Crohn's Disease. *Inflammatory bowel diseases* **24**, 1733-1744 (2018).

12. An overall schematic summarizing the mechanism would be helpful.

Response: It is included now in the revised manuscript (**new Fig. 10**).

13. The quantification panels of Fig. 2D and Fig. 2E are identical. Please correct.

Response: We apologize for the error, which has been corrected now in the revised manuscript.

14. Fig. 7 states the dose of C. Rodentium as 1×10^9 , while in the results section, the dose is given as 5×10^8 ? Please clarify.

Response: We apologize for the error. The dose should be 5×10^8 CFU/mice, which has been corrected now in the revised manuscript.

15. Fig. 8G was mentioned in the results section but G is not present in the figure. This needs to be updated.

Response: We apologize for the error. Done accordingly.

16. Please indicate the concentration of Etomoxir used.

Response: We do appreciate the reviewer's careful reading through our manuscript. See response to #10. We have removed all the data regarding metabolism including this one in the revised manuscript.

17. ILCs are innate lymphoid cells and not innate lymphocyte cells.

Response: Thanks for the reviewer's careful reading through our manuscript. We have corrected it.

18. Please use the correct formatting for knockout mice e.g. italics for mouse genes and superscript for flox.

Response: Done accordingly.

19. Some subtitles are needed for the figure panels to orient the reader. For example, Fig.1A and Fig.1E are so similar, readers cannot easily discern the difference between T cells and Th1 Cells as labeled.

Response: Thanks for the reviewer's excellent suggestion. Done accordingly.

20. *Figure legends for Fig. S1 (F-G) are the same as Fig. S1 (A-B).*

Response: We have been corrected the figure legends in the revised manuscript.

21. *“One representative of three independent experiments was shown” is wrote in every figure legend. Is the true for each Figure?*

Response: Thanks for the reviewer’s careful reading through our manuscript. We did every experiment at least three times except the following data. 1) We used three samples per group for RNAseq analysis, which were done only one time. 2) The data in revised Figure. 6D-E were one representative data of two independent experiments, which we wrote “One representative of two or three independent experiments was shown” in the original Figure 5 legend. To make the statement more clearly, we have modified the statement to “One representative of two independent experiments was shown (D and E), and One representative of three independent experiments was shown (B-C).” 3) Peripheral blood CD4⁺ T cells were isolated from 8 healthy controls, and 10 CD patients, and 7 UC patients (new Figure 9A-E). 4) The new data in revised Figure 7 G-H were one representative data of two independent experiments. We have modified the statement in Figure legends.

22. *Gating strategy for the FACS data is necessary.*

Response: Done accordingly. Please see **fig. S15**.

Reply to comments of Reviewer #2

1. *Individual experiments with only 3 replicates appear low for many of the studies.*

Response: We do appreciate the reviewer’s excellent comments. We agree with the reviewer that 3 replicates are in the lower end but they are statistically acceptable and have been fairly used in many publications, including NCOMMS. In addition, the cells we used *in vitro* cultures are almost identical in each experiment as they are from the same mice and being isolated from the same procedure. The varieties are very small in the results from each well, we thus trust the 3 replicates are statistically acceptable.

2. To accompany Fig 1 a list of the top 100 or so induced and repressed genes and their levels of induction/repression would be helpful to the reader.

Response: We do appreciate the reviewer's excellent comments. We now attached all the genes significantly changed in the Source file.

3. In Fig 2B/C it should be "(pg/ml)" instead of "(pg)".

Response: We appreciate the reviewer's careful reading through our manuscript, which has been corrected now in the revised manuscript.

3. In Fig 3D/4A/B please label (or reference in the legend/text) the MW of the proteins.

Response: We appreciate the reviewer's suggestion. Done accordingly.

4. How are SCFAs directly sensed by cells to inhibit HDAC activity if not via GPR43?

Response: See response to Reviewer #1 question #4 and #9. In addition to activation of G-protein-coupled receptors (GPCRs), SCFA can enter into cells through passive diffusion or carrier-mediated transportation to function as HDAC inhibitors to regulate cell differentiation and function. In our study, we found butyrate promoted IL-22 production through their inhibition on HDAC activity but not GPR43. Following the reviewer's suggestion, we now investigated whether GPR41 and GPR109a, the other two receptors for SCFA, mediate the butyrate induction IL-22 in CD4⁺ T cells and ILCs. We found that similar to butyrate, GPR41 specific agonist promoted IL-22 production in both CD4⁺ T cells and ILCs. However, deficiency in CD109a did not affect butyrate induction IL-22, indicating that in addition to inhibition of HDAC, butyrate induction of IL-22 is also GPR41 dependent, at least partially. These data are included in the revised manuscript now (**new Fig. 3A-D, fig. S8, S9A-C**). Furthermore, HDAC inhibitor and GPR41 agonist cooperatively promoted IL-22 production in T cells, in which IL-22 levels were similar as induced by butyrate (**new Fig. 3A-C**).

5. Data in Fig 6 showing that SCFAs enhance oxygen consumption and glycolysis does not appear well integrated into the rest of the manuscript. Additional rationale, discussion appears warranted.

Response: We appreciate the reviewer's excellent comments. According to the comments of Reviewer #1 and Reviewer #2, we removed all the data regarding metabolism in the revised manuscript.

6. *In Fig 6 D is should be "ECAR" not "EACR".*

Response: We do appreciate the reviewer's careful reading through our manuscript. See response to #5. We have removed all the data regarding metabolism including this one in the revised manuscript.

7. *In the abstract and main text it is stated that SCFAs promote T cell IL-22 production from patients with Crohn's disease. This is technically correct, but SCFAs also promote T cell IL-22 production from HC. Conclusions should be modified accordingly.*

Response: We do appreciate the reviewer's excellent comments. We have modified the conclusion to "SCFAs promote human T cell IL-22 production" accordingly in the revised manuscript.

8. *Throughout the manuscript the authors refer to "T cells" when then are assessing CD4+ T cells. It is recommended that they are referred to as "CD4+ T cells" to be accurate.*

Response: We do appreciate the reviewer's professional suggestion. We have done accordingly.

Reply to comments of Reviewer #3

1. *The premise of the paper is that SCFAs promote IL-22 production from CD4 T cells and ILCs to protect against intestinal inflammation. If that's the case why are splenocytes chosen over cells from the GALT or lamina propria? Wouldn't MLN also be a more appropriate source for cell isolation compared to the spleen? Do cells from these different sites respond differently to the SCFA? Do they polarize differently? Do they proliferate at different rate?*

Response: We do appreciate the reviewer's outstanding comments. The thrust of this study is to investigate whether SCFAs promote T cell and ILC production of IL-22 to contribute to the maintenance of intestinal homeostasis and protect the intestines from inflammation. As we are investigated the basic mechanisms which would apply for all CD4 T cells and ILCs, and because the high yields of spleen cells, we used splenic cells *in vitro* experiment. As suggested by the

reviewer, we also repeated the experiments using MLN CD4⁺ T cells to verify the butyrate induction of IL-22 in CD4⁺ T cells, and intestinal propria cells to confirm the effect of butyrate in ILCs. Consistent with splenic CD4⁺ T cells and ILCs, butyrate promoted IL-22 production in MLN CD4⁺ T cells and lamina propria ILCs. Additionally, splenic and MLN CD4⁺ T cells are polarizing and proliferating at similar levels under Th1, Th17, and Treg conditions. These data are included in the revised manuscript now (**new fig. S3**).

2. While I understand the reason for testing the CBir1 transgenics, they do pose problems. Having a transgenic TCR completely changes how thymocytes develop and what central and peripheral tolerance mechanisms are in play. The transgenic TCR in question is specific to commensal flagellin, which is abundant in the GI tract. Are these mice germ-free? If not, the T cells would likely have already encountered their cognate antigen and undergone changes such as deletion, anergy, or worse epigenetic reprogramming (Greenberg, 2012). I think it would be better for the authors to focus on WT cells, but examine them from different sites, including the gut.

Response: We generated CBir1 Tg mice several years ago [1], which are specific for gut microbiota antigen CBir1 flagellin. Although gut microbiota antigen CBir1 flagellin is abundant in GI track as the reviewer correctly pointed out, the T cells from CBir1 Tg mice respond to CBir1 antigen stimulation very well *in vitro*, they do not proliferate *in vivo* when transferred into wild type mice which contain high levels of commensal CBir1 antigens in the intestines. However, these CBir1 Tg T cells are not anergic as evidenced by the facts that the CBir1 Tg T cells re-isolated from the recipient mice proliferate well when re-stimulated *in vitro* with CBir1 antigen. These CBir1 Tg CD4 T cells are neither tolerized as they do respond to CBir1 flagellin administered I.P. *in vivo*. Interestingly, they proliferate strongly once transferred into IgA KO mice, indicating that intestinal IgA blocks the gut microbiota antigen to stimulate CBir1 T cells *in vivo*. Thus, the CBir1 T cells have not already encountered their cognate CBir1 antigen in the intestines, this have not undergone deletion or anergy [1]. We thus used CBir1 Tg T cells in this study as they are gut microbiota antigen specific and which cause colitis in animals [2]. To follow the reviewer's point, we have also used T cells from wild type B6 mice in this study using anti-CD3 and anti-CD28 mAb stimulation, and obtained the same results as using CBir1 T cells. Some of the data have been shown in the manuscript.

- [1] Cong Y, Feng T, Fujihashi K *et al.* A dominant, coordinated T regulatory cell-IgA response to the intestinal microbiota. *Proceedings of the National Academy of Sciences of the United States of America* **106**, 19256-19261 (2009).
- [2] Feng T, Wang L, Schoeb TR *et al.* Microbiota innate stimulation is a prerequisite for T cell spontaneous proliferation and induction of experimental colitis. *The Journal of experimental medicine* **207**, 1321-1332 (2010).

3. *In this report, an epigenetic mechanism driven by Hif1a was proposed. If this is the case, then perhaps the effect of oxygen tension on the capacity of T cells to produce IL-22 should be taken into account, and only T cells from the GALT should be used instead of the spleen? Cladwell et al (Cladwell, 2011) seem to suggest that spleen is rather hypoxic compare to the GALT.*

Response: We do appreciate the reviewer's excellent point regarding the effect of oxygen tension on T cell IL-22 production. See response to question #1. As suggested, we also used MLN CD4 T cells in vitro studies and measured IL-22 production in lamina propria T cells and ILCs after treatment with butyrate *in vivo*. Butyrate upregulated IL-22 production similarly in both splenic and MLN CD4⁺ T cells (**new fig. S3**).

4. *As for the Th polarizing conditions, does the addition of anti IFN γ /IL-4 cytokine antibodies affect the outcome of polarization and IL-22 production? Does the addition of IL-23 affect Th17/ILC polarization/response and IL-22 secretion? IL-2 is critical for several lineages of T cells, it is very surprising to see that it's not mentioned even once in this article.*

Response: We do appreciate the reviewer's excellent comments, and apologize that we did not write the T cell proliferation protocol in more details. CD4⁺ T cells were activated with anti-CD3/28 under neutral, Th1 (10 ng/ml IL-12), Th17 (15 ng/ml TGF β , 30 ng/ml IL-6, 10 μ g/ml anti-IFN γ mAb, 5 μ g/ml anti-IL-4 mAb), or Treg (5 ng/ml TGF β and 10 μ g/ml anti-IFN γ mAb) polarization conditions. Additionally, we found anti-IFN γ /IL-4 promoted Th17 polarization, but did not affect IL-22 production (**A**). IL-23 mildly increased Th17 polarization and IL-22 production, but dramatically upregulated IL-22 production in ILCs (**A, D**). IL-22, which is critical for T cell proliferation, can be secreted by T cells after activation. Since IL-2 suppress Th17 differentiation, we checked whether IL-2 affects Th1/ Treg polarization and IL-22

production. We found addition of IL-2 did not affect Th1 and Treg polarization, as well as IL-22 production (B-C).

5. Please provide more detail on how the T cells were expanded (ie, how many were activated, what CD3/CD28 clones were used in what kind of plates/culturing container, at what cellular concentration/number before during and after the expansion). Also, please provide some CD44 stain data on the T cells before and after the expansion. These data would confirm that the T

cells are viable and hence the differential cytokine expression seen is purely due to the presence of butyrate.

Response: We appreciate the reviewer's comments. 0.2 million/ml CD4⁺ T cells were activated with 5 µg/ml αCD3 mAb (Clone# 145-2C11, Bio X Cell) and 2µg/ml αCD28 mAb (Clone# 37.51, Bio X Cell) plated in the 24-well plates at 37°C with 5% CO₂. On day 5, the cells were harvested (around 2 million/ml) for analysis the cytokine expression. As suggested, the cell activation was checked before and after activation.

CD4⁺ T cells were activated with anti-CD3/28 for 5 days. Cell activation was assessed by CD25 and CD44 using flow cytometry.

6. Figure 2. Dendritic cells are an important source of IL-23 and play critical roles in the polarization of T helper cells. Are there any changes in IL-23 secretion by the dendritic cells in the MLN and lamina propria after 3 weeks of butyrate in drinking water?

Response: We do appreciate the reviewer's excellent point, and have done accordingly. Butyrate did not affect IL-23 production in dendritic cells in the MLN and lamina propria *in vivo*. Please see the **new fig. S5B**.

7. Also, please provide data on the intestinal microbiota. Butyrate can modulate oxygen availability, and since the intestinal microbiota can shift drastically based on oxygen availability. It would be helpful to determine whether the change in IL-22 production is due to butyrate, the change of microbiota or a combination of both.

Response: We do appreciate the reviewer's excellent thoughts. We have shown previously that butyrate affected intestinal microbiota in mice [1]. To follow the reviewer's point, we measured butyrate levels in the colons before and after *in vivo* administration of butyrate, and found that administration of butyrate increased the butyrate levels in colon (**revised Fig. 2B**). Therefore, as suggested by the reviewer, butyrate induction of IL-22 *in vivo* is likely due to the combination of altered microbiota and increased butyrate, which has been included in the discussion section of the revised manuscript.

- [1] Zhao Y, Chen F, Wu W *et al.* GPR43 mediates microbiota metabolite SCFA regulation of antimicrobial peptide expression in intestinal epithelial cells via activation of mTOR and STAT3. *Mucosal immunology* **11**, 752-762 (2018).

8. Downregulation of CD4 or CD8 can be a sign of T cell activation. How are these T cells gated in figure 2D? What would the data look like if the plot were not gated on the CD4 high cells first? Are there any CD4⁻ cells secreting IL-22 in these sites? What are the absolute numbers for these IL-22 secreting cells?

Response: We appreciate the reviewer's excellent comments. Live single CD4⁺ T cells were gated in Figure 2D. We also included the absolute numbers (% IL-22⁺ CD4⁺ T cells × Total spleen or MLN or lamina propria cells) in the revised manuscript (**revised Fig. 2E-H**). CD4⁻ populations also produced IL-22 production. (see **new fig. S6**).

9. IL-22 can be secreted by NKT cells, which can be protective in intestinal inflammation. Does butyrate enhances IL-22 secretion from NKT cells as well?

Response: As suggested, we measured NKT cell IL-22 production, and found that butyrate did not affect IL-22 production in NKT cells. This data is included in the revised manuscript now (**new fig. S5A**).

10. Figure 3. Is butyrate activating Hif1a? Can butyrate activate AHR? There seem to be evidence that butyrate can activate AHR in human intestinal epithelial cells (Marinelli, 2019), is this true in mice as well?

Response: We appreciated the reviewer's excellent points. As suggested, we used Xenobiotic Response Element (XRE/AhR) Luciferase Reporter Gene Lentivirus to analyze whether butyrate activates AHR in mice cells, and found that butyrate indeed activated AHR. Please see revised Methods part in supplementary and **revised Fig. 4G**.

11. What is the rationale of YC-1 as Hif1a inhibitor? There were reports that YC-1 also acts on Hif2a. Will other Hif1a inhibitors produce the same affect? Although Hif1a expression was shown, what about its activity? How much Hif1a activity is YC-1 blunting? How about Hif2a?

For figure 3D, why not show both Hif1a Hif2a and AhR western blot? DMOG treated T cells would be a nice control to have (please see reference 34).

Response: We appreciate the reviewer’s careful reading through our manuscript. According published papers, YC-1 is widely used as HIF1 α inhibitor, although it might affect HIF2 α . Additionally, we used HIF1 $\alpha^{-/-}$ T cells to confirm whether HIF1 α is involved in butyrate induction of IL-22 (**new Fig. 4K**). Therefore, we chose YC-1 as HIF1 α inhibitor in this study. As suggested by the reviewer, we also used another selective HIF1 α inhibitor, FM19G11, and found FM19G11 also suppressed IL-22 production induced by butyrate in T cells (**new fig. S10B**). However, TC-S 7009, a selective inhibitor for HIF2 α , did not affect T cell production of IL-22 production (**new fig. S10C**). DMOG, which acts to stabilize HIF-1 α expression, promoted IL-22 production in T cells (**new fig. S10B**). We also showed that butyrate upregulated HIF1 α , AHR, but not HIF2 α protein expression in T cells (**revised fig. 4D-F, new fig. S10A**). All these data are included in the revised manuscript.

12. Please provide the rationale why the Cbir1 transgenic were used in one part of the figure and B6 for the other part? According to the methods section, they were all stimulated by PMA/Ionomycin, which defeats the purpose of a TCR transgenic. Were the Cbir1 cells responsive to the YSNANILSQ peptide? If so, how did they respond to different dosages of the peptide? Would be nice to see flow data on the IFN gamma and IL-22 readout.

Response: We do appreciate the reviewer’s careful reading through our manuscript. In this study, we used two ways to activate CD4⁺ T cells to determine how SCFAs regulate CD4⁺ T cell IL-22 production: 1) WT splenic CD4⁺ T cells were activated with anti-CD3 and anti-CD28 mAb; 2) Cbir Tg CD4⁺ T cells were activated with irradiated antigen-presenting cells and 1 μ g/ml Cbir1 peptide (YSNANILSQ peptide). After 5 day’s culture, cells were activated with PMA/Ionomycin for 5 hours and brefeldin for 3 hours for analysis of cytokine expression by FACS. As suggested, we used a serious dose

of YSNANILSQ peptide to activate CBir1 Tg CD4⁺ T cells under neutral condition, and found high doses of YSNANILSQ peptide promoted production of IFN- γ , but did not affect IL-22 production (A). However, lower cell viability was seen when CBir1 T cells were activated with higher dose of YSNANILSQ peptide (10 and 100 μ g/ml) (B). In this study, we used 1 μ g/ml YSNANILSQ peptide.

13. For figures 3E and F, please show flow cytometry data. Does treatment of inhibitors affect cell viability?

Response: Done accordingly. Treatment of these inhibitors at the dose used did not affect CD4⁺ T cell viability. Please see the **new fig. S10D**.

14. For figure 3G, it appears that the majority of IL-22 producing cells in the butyrate treated WT cells are low IL-22 producers. Would the inclusion of IL-22 FMO/ISO help determine whether these are indeed IL-22 secreting cells? Also, it is rather difficult to see these dots. It would be preferable to show the plots in low-resolution mode so that it's easier to see the dots but more preferable that more events were acquired (50-100,000 CD4 T cells should suffice). Also please include Hif2a^{-/-} T cells in the data if Hif2a activity is indeed affected by the inhibitor.

Response: We appreciate the reviewer's outstanding comments. As suggested, IL-22 FMO/ISO staining was included in **new fig. S7**. Per Journal policy of Nature Communications, we have changed all the FACS plots to contour plots with outliers. Since butyrate did not affect HIF2 α expression, and HIF2 α inhibitor did not affect IL-22 production neither (**new fig. S10A and S10C**), we thus did not use the HIF2 α ^{-/-} T cells.

15. The figure legend of (H-I) "...Th1 conditions with or without (0.5mM) under normxic" does not make sense, is the word missing butyrate and the typo normoxic? Further, what is the justification of using 3% oxygen (please see Cladwell et al, 2011)? Would a T cell in the lamina propria/MLN experience such concentration of oxygen (please see Carreau 2011, Espey, 2013)?

Response: We do appreciate the reviewer's excellent comments and apologize for such careless errors. We have corrected the figure legend as suggested. First, according to the reports [1, 2], the P_{O₂} in mouse intestine was around 1-40 mmHg. Standard atmospheric, sea-level, pressure is

approximately 760 mmHg, so 1-40 mmHg is 0.13-5.26% O₂. Therefore, we chose the 3% O₂, which is the average value of % O₂ in mouse intestine.

- [1] Zheng L, Kelly CJ, Colgan SP. Physiologic hypoxia and oxygen homeostasis in the healthy intestine. A Review in the Theme: Cellular Responses to Hypoxia. American journal of physiology Cell physiology **309**, C350-360 (2015).
- [2] Espey MG. Role of oxygen gradients in shaping redox relationships between the human intestine and its microbiota. Free radical biology & medicine **55**, 130-140 (2013).

16. Figure 4. For 4A&B, it would be nice to have density quantification and stats. The effect of butyrate on mTOR shown in 4B seemed minimal, would phosphor flow be a better option? For the phosphor flow data, please indicate what fluorochrome and clone in the methods section. For 4C, how were the cells stained and gated? Please provide more information in the methods section. Please also provide MFI and stats.

Response: We do appreciate the reviewer's excellent suggestion. We now analyzed the density of pStat3 and pmTOR bands, and also measured the p-Stat3 and p-mTOR by FACS as suggested by the reviewer, which are included in the revised manuscript (**revised Fig. 5A-E**). The antibody information was included in Methods part and Reporting Summary file. Cell gating Strategy was included in **fig. S15**.

17. For 4H why was there a drop in IL-22 in the WT? This look very different compared to 3G 2D 2E and 1H. Also, the figure legend for 4H is confusing. Can treatment with butyrate alone result in IL-22 production?

Response: We appreciate the reviewer's comments. That the Y-axis (IL-22-PE) value range is not same in all the FACS dots might result in the different shape in different figures, so we uniformed the Y-axis (IL-22-PE) range to 0-10⁵ in all these FACS dots. The figure legend was also corrected as suggested.

18. Figure 5. If IL-22 secretion was assessed on day 5 post activation, what is the justification of performing ChIP on IL-22 promoter on day 2?

Response: We do appreciate the reviewer's question. First, butyrate induced IL-22 expression as early as at 24 h and reached peak mRNA level at 60 h. Then, we analyzed HIF1 α expression at

24, 48, 60 h by western-blot, and found HIF1 α expressed relatively high and reached significance on day 2. CHIP was performed to check whether HIF1 α protein could bind to the *il22* promoter, thus affect IL-22 transcription. Therefore, we chose that time point for ChIP assay.

19. 5A: To make the claim that these HRE binding sites are indeed binding Hif1 α one need to first list the DNA sequence, then map for local topology, and then follow with experiments showing that Hif1 α is indeed binding on these sites.

Response: We do appreciate the reviewer's comments. As suggested, the DNA sequence is listed (**new Fig. 6**). However, due to lack of access to a software for topology, we do not have a map for local topology. We do apologize for that.

20. For the ChIP experiments, please elaborate on how the fold enrichment was calculated in your methods section and provide cytokine data for day 2.

Response: We appreciate the reviewer's comments. We apologize for the oversights. The way to calculate the fold enrichment in CHIP experiments were included in Methods section in the revised manuscript now. IL-22 levels on day 2 were showed in **Fig. 1B** in the revised manuscript.

21. Figure 6. Please show metabolism data on GPR43^{-/-} and STAT3^{-/-} CD4 T cells. The difference seen could be due to additional activation signal induced by butyrate through GPR43 and STAT3.

Response: We appreciate the reviewer's excellent suggestion. Please see responses to Reviewer #1 question #4 and Reviewer #2 question #4. As suggested from Reviewer #1 and Reviewer #2, we removed all the data related with metabolism in the revised manuscript.

22. Figure 7. For 7B, the histology shown is too small to really determine what is going on. Please include pathogen burdens. marginal, please include WT uninfected. For flow cytometry data, please provide data in numbers not percentages. The numbers of events on the plots are not even, making it difficult to analyze.

Response: We appreciate the reviewer's excellent suggestion. We have enlarged the histologic images, and included the uninfected WT mice (**revised Fig. 7B**). As we did not use fluorescence labeled *Citrobacter Rodentium* for infection, we could not count the *Citrobacter* burden in the intestines as once then intestines were digested, most bacteria died. However, we did check *Citrobacter* in livers, and the *Citrobacter* load in feces (**revised Fig. 7E-F**), which have been widely used in many laboratories for measurement the bacterial clearance. We also included the absolute numbers ($\% \text{ IL-22}^+ \text{ CD4}^+ \text{ T cells} \times \text{Total lamina propria cells}$) in the revised manuscript (**revised Fig. 7C**).

23. *Figure 8. For 8B. It is CD4 T cells that were isolated? If so, how were they stained and gated? Why choose SSC over CD4? Please provide clone names and fluorochrome for antibodies and gating strategy in figure legend or in methods. Also for the CD patients, how were they treated for their disease?*

Response: Human CD4⁺ T cells were isolated from peripheral blood using anti-human CD4 magnetic particles. After activation and culture, T cells were stained with Live/dye, CD4, and IL-22. As suggested by the reviewer, we now changed the profiles to use CD4 instead of SSC in **revised Fig. 9**. The clone names and fluorochrome for antibodies were included in methods and reporting summary file, and the gating strategy was included in the revised manuscript (**fig. S15**). As shown in **Table S1**, 3 CD patients were treated with 5-aminosalicylates, and 6 CD patients were treated with nutritional therapy. None of them received the immunosuppressants and biologics therapy.

We thank the editor and three reviewers for their careful reading of the manuscript and especially for their helpful comments, which make this manuscript much better. With the inclusion of the responses provided above and the new data, we believe that we have responded in full to the comments of the three reviewers and editor. We trust that this revised manuscript will now be suitable for publication in *Nature Communications*. Thank you for your consideration of this revised manuscript.

REVIEWERS' COMMENTS:

Reviewer #1 (Remarks to the Author):

In the revised manuscript, the authors have sufficiently addressed all our concerns, and added significant new evidence in mice and humans to support their proposed hypotheses. This is an exceptional manuscript.

Reviewer #2 (Remarks to the Author):

The authors have thoroughly address my concerns and comments and the overall revision has dramatically improved the manuscript.

Reviewer #3 (Remarks to the Author):

The authors have largely addressed my concerns.

I still have some minor suggested changes that should improve the flow of the paper.

In the interest of the flow of paper, perhaps move items such as NKTs not producing IL-22 to discussion or provide rationale and reference of why NKTs were tested. As it stands it's rather abrupt. Same goes for Hif2a.

For the ease of the reader to follow please:

Include the method of stimulation in the figure legend.

For example, line 792 flow cytometry on day 5 following PMA/ionomycin stimulation (H)

Line 103: reference might be needed.

Line 146: IL-22 instead of IL22.

Figure 7B goblet cell hyperplasia in the butyrate treated mice is striking, and should be included in the discussion as an effect of IL-22 following butyrate treatment.

Reply to comments of Reviewer #1

1. In the revised manuscript, the authors have sufficiently addressed all our concerns, and added significant new evidence in mice and humans to support their proposed hypotheses. This is an exceptional manuscript.

Response: Thanks. We appreciate the reviewer's previous excellent comments and suggestion, which made this manuscript much stronger.

Reply to comments of Reviewer #2

1. The authors have thoroughly addressed my concerns and comments and the overall revision has dramatically improved the manuscript.

Response: Thanks. We do appreciate the reviewer's previous constructive comments and suggestions.

Reply to comments of Reviewer #3

1. The authors have largely addressed my concerns. I still have some minor suggested changes that should improve the flow of the paper.

In the interest of the flow of paper, perhaps move items such as NKTs not producing IL-22 to discussion or provide rationale and reference of why NKTs were tested. As it stands it's rather abrupt. Same goes for Hif2a.

Response: We do appreciate the reviewer's careful reading through our manuscript. As suggested, the rationale and reference of NKT and Hif2a were included in the revised manuscript.

2. For the ease of the reader to follow please:

Include the method of stimulation in the figure legend.

Response: Due to the word limitation of figure legends, we included the stimulation methods in the Methods section.

*3. For example, line 792 flow cytometry on day 5 following PMA/ionomycin stimulation (H)
Line 103: reference might be needed.*

Response: Done accordingly.

4. *Line 146: IL-22 instead of IL22.*

Response: corrected.

5. *Figure 7B goblet cell hyperplasia in the butyrate treated mice is striking, and should be included in the discussion as an effect of IL-22 following butyrate treatment.*

Response: This point has been discussed in discussion section of the revised manuscript.

We thank the editor and three reviewers for their careful reading of the manuscript and especially for their helpful comments, which make this manuscript much better. We trust that this revised manuscript will now be suitable for publication in *Nature Communications*. Thank you for your consideration of this revised manuscript.